# 2 OLMo 2 Furious (COLM's Version)

**OLMo Team**[*]

**Pete Walsh**[♥,1]   **Luca Soldaini**[♥,1]   **Dirk Groeneveld**[♥,1]   **Kyle Lo**[♥,1]

**Shane Arora**[♥,1]   **Akshita Bhagia**[♥,1]   **Yuling Gu**[♥,1]   **Shengyi Huang**[♥,1]
**Matt Jordan**[♥,1]   **Nathan Lambert**[♥,1]   **Dustin Schwenk**[♥,1]   **Oyvind Tafjord**[♥,1]

**Taira Anderson**[1]   **David Atkinson**[1]   **Faeze Brahman**[1]   **Christopher Clark**[1]
**Pradeep Dasigi**[1]   **Nouha Dziri**[1]   **Allyson Ettinger**[1]   **Michal Guerquin**[1]
**David Heineman**[1]   **Hamish Ivison**[1,2]   **Pang Wei Koh**[1,2]   **Jiacheng Liu**[1,2]
**Saumya Malik**[1]   **William Merrill**[1,3]   **Lester James V. Miranda**[1]   **Jacob Morrison**[1]
**Tyler Murray**[1]   **Crystal Nam**[1]   **Jake Poznanski**[1]   **Valentina Pyatkin**[1,2]
**Aman Rangapur**[1]   **Michael Schmitz**[1]   **Sam Skjonsberg**[1]   **David Wadden**[1]
**Christopher Wilhelm**[1]   **Michael Wilson**[1]   **Luke Zettlemoyer**[2]

**Ali Farhadi**[1,2]   **Noah A. Smith**[♥,1,2]   **Hannaneh Hajishirzi**[♥,1,2]

[1] Allen Institute for AI   [2] University of Washington   [3] New York University

## Abstract

We present OLMo 2, the next generation of our fully open language models. OLMo 2 includes a family of dense autoregressive language models at 7B, 13B and 32B scales with fully released artifacts—model weights, full training data, training code and recipes, training logs and thousands of intermediate checkpoints. In this work, we describe our modified model architecture and training recipe, focusing on techniques for achieving better training stability and improved per-token efficiency. Our updated pretraining data mixture introduces a new, specialized data mix called DOLMINO MIX 1124, which significantly improves model capabilities across many downstream task benchmarks when introduced via late-stage curriculum training (i.e. specialized data during the annealing phase of pretraining). Finally, we incorporate best practices from Tülu 3 to develop OLMo 2-INSTRUCT, focusing on permissive data and extending our final-stage reinforcement learning with verifiable rewards (RLVR). Our OLMo 2 base models sit at the Pareto frontier of performance to training compute, often matching or outperforming open-weight only models like Llama 3.1, Qwen 2.5, and Gemma 2 while using fewer FLOPs and with fully transparent training data, code, and recipe. Our fully open OLMo 2-INSTRUCT models are competitive with open-weight only models of comparable size and even some proprietary models like GPT-3.5 Turbo and GPT 4o Mini.

🤗 allenai/olmo-2   ⬤ allenai/olmo-core   ✣ allenai.org/olmo

## 1 Introduction

The open language model ecosystem has grown rapidly in the past year. We've seen a surge in open weights models from established developers—Llama 3 (Grattafiori et al., 2024),

---

[*] OLMo was a team effort. [♥]marks core contributors; see Author Contributions for full details. Contact olmoteam@allenai.org.

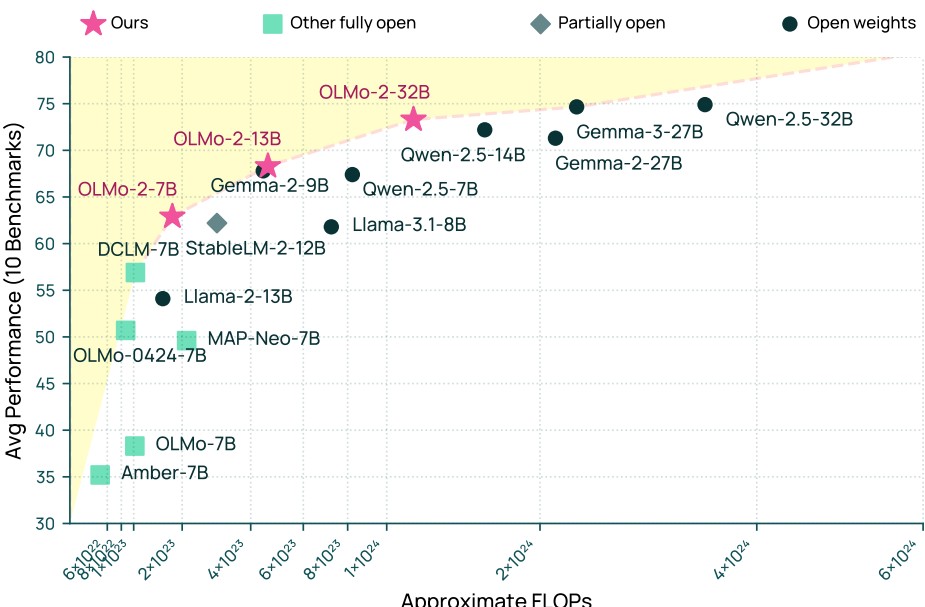

Figure 1: Performance to pretraining FLOPs (≈ 6 × training tokens × model size; Kaplan et al., 2020) for OLMO 2 and comparable models. We see that the fully open OLMO 2 lies on the Pareto frontier of training efficiency, competitive with other models of varying levels of openness at multiple sizes. For full results, see Table 3.

DBRX (Databricks, 2024), Yi 1.5 (Young et al., 2024), Qwen 2 (Yang et al., 2024a), Falcon (TII, 2024a;b), Mistral (Mistral, 2024a), Ministral (Mistral, 2024b), Phi (Abdin et al., 2024a;b)—and new contributors— Gemma (Gemma Team et al., 2024a;b; Team et al., 2025), Grok (X.AI, 2023), Command R (Cohere, 2024a;c;b) —substantially closing the gap between publicly available and closed systems (Cottier et al., 2024). Yet, these open-weights models are only the *final* artifacts of sophisticated language model recipes and complex development pipelines, and by themselves are not sufficient to support diverse forms of research into language model behaviors and uses.

In response, prior works including our first OLMO (Groeneveld et al., 2024), Pythia (Biderman et al., 2023), Amber (Liu et al., 2023c), DCLM (Li et al., 2024), MAP Neo (Zhang et al., 2024a) and SmolLM (Allal et al., 2024a;b) have adopted a **fully open approach**, releasing not just model weights but also training data, training code and well-documented recipes to support reproduction. Artifacts from fully open language modeling efforts have played a crucial role in studying training dynamics (Land & Bartolo, 2024; Jin & Ren, 2024), concept acquisition (Chang et al., 2024), and memorization (Antoniades et al., 2024; Shaib et al., 2024) in language models. Despite these developments, a gap remains between the models with the best reported performance and that of open models.

We introduce **OLMO 2**, a new family of fully open 7B, 13B and 32B models trained on up to 6T tokens. On English academic benchmarks, these models are competitive with the open weight Llama 3.1, Qwen 2.5, and Gemma 2 families of models (Figure 1). We further validate our pretrained model is an effective base model for downstream post-training by applying our Tülu 3 recipe (Lambert et al., 2024). The resulting family of models, called OLMO 2-INSTRUCT, are competitive with powerful open-weights only models and even some popular proprietary models like GPT-3.5 Turbo and GPT 4o Mini. In addition to describing our full modeling, training, and data recipes, we focus on two key areas that proved critical during the development of OLMO 2—**pretraining stability** (§4) and **mid-training recipes** (§5). Finally, we release all model weights, data, training and evaluation code, intermediate checkpoints, and recipes openly for the broader research community. A full version of our work, including additional details about post-training and our training infrastructure, can be found at OLMo et al. (2024).

## 2 The OLMO 2 Family

**Model Architecture** We adopt a decoder-only transformer architecture based on Vaswani et al. (2017), and deliver 7B, 13B and 32B parameter variants (Table 1) Our architecture is very similar to OLMO 1 (Groeneveld et al., 2024) and OLMO-0424 (Ai2, 2024) and make modifications aimed at improving training stability and performance. For space considerations, we present a full list of architecture departures in Appendix E, and discuss several key interventions in §4. Throughout this work, we also reference a 1B parameter variant used for ablation studies that inform development of our larger model targets; its details are in Appendix B.

**Tokenizer** While OLMO 1 and OLMO-0424 use the GPT-NeoX-20B tokenizer vocabulary (Black et al., 2022), we adopt c1100k, which was developed for GPT-3.5 (OpenAI, 2023a) and GPT-4 (OpenAI, 2023b). The larger vocabulary is more suited to the parameter count of the OLMO 2 family (Tao et al., 2024). In an ablation study with 1B models trained to 100B tokens, the new vocabulary slightly improves performance on a suite of downstream tasks (+0.3 to +0.8 points); further details in Appendix §C.

|  | OLMO 2 7B | OLMO 2 13B | OLMO 2 32B |
|---|---|---|---|
| **Layers** | 32 | 40 | 64 |
| **Hidden Size** ($d_{model}$) | 4096 | 5120 | 5120 |
| **Attention Heads (Q/KV)** | 32/32 (MHA) | 40/40 (MHA) | 40/8 (GQA) |
| **Batch Size** | 1024 | 2048 | 2048 |
| **Sequence Length** | 4096 | 4096 | 4096 |
| **Gradient Clipping** | 1.0 | 1.0 | 1.0 |
| **Peak LR** | $3.0 \cdot 10E-4$ | $9.0 \cdot 10E-4$ | $6.0 \cdot 10E-4$ |
| **LR Warmup** | 2000 steps | 2000 steps | 2000 steps |
| **LR Schedule (Cosine)** | 5T tokens | 5T tokens | 6.5T tokens |
| **LR Schedule Truncation** | *(after 4T)* | n/a | *after 6T* |

Table 1: OLMO 2 hyperparameters.

### 2.1 Training OLMO 2

Following recent advances in curriculum learning (Blakeney et al., 2024; Ibrahim et al., 2024; Feng et al., 2024), OLMO 2 models are trained in **two stages**, each with its corresponding data mix. In total, OLMO 2 7B is trained on 4.05 trillion tokens (3.90 trillion for pretraining stage), OLMO 2 13B is trained on 5.6 trillion tokens (5 trillion for pretraining stage), and OLMO 2 32B is trained on 6.6 trillion tokens (6.06 trillion for pretraining stage).

#### 2.1.1 Stage 1: Pretraining

**Training** The first stage—*pretraining*—is the longest (90–95% of training FLOPs). We report key architecture and training details in Table 1. Key details include our switch from multi-head attention (MHA) to grouped query attention (GQA) (Ainslie et al., 2023) to scale the 32B model, inspired by its use in concurrent work Qwen 3 (Yang et al., 2025). OLMO 2 training used random initialization from a truncated normal distribution with a mean of 0 and a standard deviation of 0.02 and a learning rate schedule that warms up the learning rate from 0 to the peak learning rate over 2000 steps, followed by a cosine decay calibrated to reach 10% of the peak learning rate after a specified max tokens. We describe these and other methods in detail in §4.

**Data** We adopt the pretraining data used in OLMOE (Muennighoff et al., 2024)—a mix of documents from DCLM (Li et al., 2024) (Common Crawl), Dolma 1.7 (Soldaini et al., 2024) (knowledge-rich documents from scientific papers, Wikipedia, etc.), and StarCoder (Li et al., 2023) (code repositories). This data mix consists of ≈ 3.9 trillion tokens. We provide a full breakdown in Appendix D.

### 2.1.2 Stage 2: Mid-training

**Training** We refer to the shorter second stage as *mid-training* (5–10% of training FLOPs), where we linearly decay the learning rate to zero over the remaining length of the run.[1]

**Data** We curated a smaller, focused mixture—**DOLMINO MIX 1124**—to imbue the model with domain knowledge from increased exposure to STEM references and high quality text as well as skills that remained lacking after the initial pretraining stage (e.g. math-solving capabilities). We up-sample high-quality web documents and curated non-web sources; we also employ synthetic data crafted to patch math capabilities of the model. We present a full breakdown of DOLMINO MIX 1124 sources in Table 6 and discuss details in §5.2.

**Model Merging or "Souping"** To get the most out of this high-quality data, and to find a better local minimum, we perform this step multiple times with different random data orders, and then average the resulting models (Matena & Raffel, 2022; Wortsman et al., 2022). For OLMo 2 7B, we anneal three separate times for 50B tokens each, with different randomized data orders; we average the resulting models to produce the final model. For both OLMo 2 13B and OLMo 2 32B, we train three separate times for 100B tokens each (same number of update steps as the 7B), and then a fourth time for 300B tokens. The final model is the average of all four models. Table 14 in Appendix §H summarizes data composition of the 50B, 100B and 300B sets.

**Overall** Despite minimal compute, mid-training provides a significant downstream performance boost to a pretrained base model: +18.7% for the 7B model, +15.9% for the 13B model, and +12.3% for the 32B model; see Table 2. To ensure we aren't harming our base model's potential for post-training, we also train and evaluate OLMo 2-INSTRUCT using our Tülu 3 post-training recipe (see §3).

| Model | Stage | Avg | Dev Benchmarks | | | | | | Held-out Evals | | | |
|---|---|---|---|---|---|---|---|---|---|---|---|---|
| | | | MMLU | ARC$_C$ | HS | WG | NQ | DROP | AGI | GSM | MMLU$_P$ | TQA |
| 7B | 1 | 53.0 | 59.8 | 72.6 | 81.3 | 75.8 | 29.0 | 40.7 | 44.6 | 24.1 | 27.4 | 74.6 |
| | 2 | 62.9 | 63.7 | 79.8 | 83.8 | 77.2 | 36.9 | 60.8 | 50.4 | 67.5 | 31.0 | 78.0 |
| 13B | 1 | 58.9 | 63.4 | 80.2 | 84.8 | 79.4 | 34.6 | 49.6 | 48.2 | 37.3 | 31.2 | 80.3 |
| | 2 | 68.3 | 67.5 | 83.5 | 86.4 | 81.5 | 46.7 | 70.7 | 54.2 | 75.1 | 35.1 | 81.9 |
| 32B | 1 | 64.9 | 72.9 | 88.7 | 86.5 | 82.4 | 40.6 | 57.3 | 56.8 | 56.2 | 42.0 | 85.5 |
| | 2 | 72.9 | 74.9 | 90.4 | 89.7 | 83.0 | 50.2 | 74.3 | 61.0 | 78.8 | 46.9 | 88.0 |

Table 2: Impact of our mid-training recipe on downstream tasks.

## 3 Evaluation and Results

OLMo 2 is evaluated via standard language model benchmarks. Further, we apply post-training to OLMo 2 and evaluate the result—OLMo 2-INSTRUCT—on a diverse set of tasks to assess the adaptation potential of our base model.

**Base Model Evaluation:** We evaluated OLMo 2 and other baseline models using the OLMES evaluation suite (Gu et al., 2024), which includes a range of benchmark datasets for both multiple-choice and generative tasks, using standardized prompts and in-context examples for few shot predictions. Full descriptions of benchmark tasks in Appendix A.1. For multiple-choice tasks, we evaluate accuracy; for generative tasks, we evaluate F1 to account for partial matches. Additionally, to avoid overfitting our recipe to these benchmarks, we maintained a **held-out suite of tasks** which were not used for model development decisions;

---

[1]While the concept of multiple stages of self-supervised training is not new (*e.g.*, Gururangan et al. 2020), we adopt the term *mid-training* from Abdin et al. (2024a) and OpenAI (2024).

| Model | Avg | FLOPs | Dev Benchmarks | | | | | | Held-out Evals | | | |
|---|---|---|---|---|---|---|---|---|---|---|---|---|
| | | | MMLU | ARC$_C$ | HS | WG | NQ | DROP | AGI | GSM | MMLU$_P$ | TQA |
| **Open-weights models 7-14B Parameters** | | | | | | | | | | | | |
| Llama 3.1 8B | 61.8 | 7.2 | 66.9 | 79.5 | 81.6 | 76.6 | 33.9 | 56.4 | 51.3 | 56.5 | 34.7 | 80.3 |
| Qwen 2.5 7B | 67.4 | 8.2 | 74.4 | 89.5 | 89.7 | 74.2 | 29.9 | 55.8 | 63.7 | 81.5 | 45.8 | 69.4 |
| Gemma 2 9B | 67.8 | 4.4 | 70.6 | 89.5 | 87.3 | 78.8 | 38.0 | 63.0 | 57.3 | 70.1 | 42.0 | 81.8 |
| Llama 2 13B | 54.1 | 1.6 | 55.7 | 67.3 | 83.9 | 74.9 | 38.4 | 45.6 | 41.5 | 28.1 | 23.9 | 81.3 |
| Qwen 2.5 14B | 72.3 | 16.0 | 79.3 | 94.0 | 94.0 | 80.0 | 37.3 | 51.5 | 71.0 | 83.4 | 52.8 | 79.2 |
| **Open-weights models 24-32B Parameters** | | | | | | | | | | | | |
| Gemma 2 27B | 71.3 | 21.0 | 75.7 | 90.7 | 88.4 | 74.5 | 44.7 | 70.1 | 61.5 | 75.7 | 44.7 | 87.4 |
| Qwen 2.5 32B | 74.9 | 16.0 | 83.1 | 95.6 | 96.0 | 84.0 | 37.0 | 53.1 | 78.0 | 83.3 | 59.0 | 79.9 |
| Gemma 3 27B | 74.7 | 23.0 | 79.5 | 93.4 | 88.2 | 75.0 | 45.4 | 73.2 | 69.5 | 80.4 | 52.9 | 89.1 |
| **Fully-open models** | | | | | | | | | | | | |
| Amber 7B | 35.2 | 0.5 | 24.7 | 44.9 | 74.5 | 65.5 | 18.7 | 26.1 | 21.8 | 4.8 | 11.7 | 59.3 |
| OLMo 1 7B | 38.3 | 1.0 | 28.3 | 46.4 | 78.1 | 68.5 | 24.8 | 27.3 | 23.7 | 9.2 | 12.1 | 64.1 |
| MAP Neo 7B | 49.6 | 2.1 | 58.0 | 78.4 | 72.8 | 69.2 | 28.9 | 39.4 | 45.8 | 12.5 | 25.9 | 65.1 |
| OLMo-0424 7B | 50.7 | 1.0 | 54.3 | 66.9 | 80.1 | 73.6 | 29.6 | 50.0 | 43.9 | 27.7 | 22.1 | 58.8 |
| DCLM 7B | 56.9 | 1.0 | 64.4 | 79.8 | 82.3 | 77.3 | 28.8 | 39.3 | 47.5 | 46.1 | 31.3 | 72.1 |
| StableLM 2 12B | 62.2 | 2.9 | 62.4 | 81.9 | 84.5 | 77.7 | 37.6 | 55.5 | 50.9 | 62.0 | 29.3 | 79.9 |
| OLMo 2 7B | 62.9 | 1.8 | 63.7 | 79.8 | 83.8 | 77.2 | 36.9 | 60.9 | 50.4 | 67.5 | 31.0 | 78.0 |
| OLMo 2 13B | 68.3 | 4.6 | 67.5 | 83.5 | 86.4 | 81.5 | 46.7 | 70.7 | 54.2 | 75.1 | 35.1 | 81.9 |
| OLMo 2 32B | 73.3 | 13.0 | 74.9 | 90.4 | 89.7 | 78.7 | 50.2 | 74.3 | 61.0 | 78.8 | 46.9 | 88.0 |

Table 3: OLMO 2 vs. comparable models (size, architecture) with known pretraining FLOPs, which are reported relative to 10E23. For example, OLMO 2 7B training took $1.8 \cdot 10E23$ FLOPs.

we advocate for a standard practice of declaring development vs held-out evaluation tasks for model developers.[2]

Table 3 contains overall results. We find our **OLMO 2 models are competitive with the best open-weights models** of comparable size, despite OLMO 2 requiring **far fewer training FLOPs** (see Figure 1) and maintaining **full openness (e.g. training data)**. We find that gains observed on development metrics largely translate to our unseen evaluation suite, indicative of a generalizable training recipe. Curiously, while we've found our recipe developed using 1B model ablations has generalized well to the 7B, 13B and 32B scales, our recipe may not be optimal for training smaller models (even 1B scale); we discuss this limitation in the Appendix I.

**Post-Training Recipe and Evaluation** For post-training we apply our Tülu 3 (Lambert et al., 2024) recipe with supervised finetuning, on-policy preference tuning, and reinforcement learning with verifiable rewards (RLVR).[3] The resulting models—OLMO 2-INSTRUCT—are evaluated in Table 4 on general and precise instruction following, math, knowledge

---

[2]GSM8k (Cobbe et al., 2021) was only partially held-out, as we subsampled 200 of 1319 GSM8k examples for mid-training data development when we noticed poor math capabilities after pretraining; we call this dev set GSM*. The remaining 1119 GSM8k examples we reserve as held-out and report final performance on them only.

[3]We made minor modifications to the preference data to use generations from permissively-licensed models and added a multi-stage RLVR training protocol to optimize final performance, but otherwise followed the recipe as-is.

| Model | Avg | AE2 | BBH | DROP | GSM | IFE | MATH | MMLU | Safety | PQA | TQA |
|---|---|---|---|---|---|---|---|---|---|---|---|
| Closed API models | | | | | | | | | | | |
| GPT-3.5 Turbo 0125 | 60.5 | 38.7 | 66.6 | 70.2 | 74.3 | 66.9 | 41.2 | 70.2 | 69.1 | 45.0 | 62.9 |
| GPT 4o Mini 0724 | 65.7 | 49.7 | 65.9 | 36.3 | 83.0 | 83.5 | 67.9 | 82.2 | 84.9 | 39.0 | 64.8 |
| Open weights models 7-14B Parameters | | | | | | | | | | | |
| Llama 3.1 8B | 59.1 | 25.8 | 71.9 | 61.7 | 83.4 | 80.6 | 42.5 | 71.3 | 70.2 | 28.4 | 55.1 |
| Gemma 2 9B | 58.1 | 43.7 | 64.9 | 58.8 | 79.7 | 69.9 | 29.8 | 69.1 | 75.5 | 28.3 | 61.4 |
| Qwen 2.5 7B | 61.6 | 29.7 | 70.2 | 54.4 | 83.8 | 74.7 | 69.9 | 76.6 | 75.0 | 18.1 | 63.1 |
| Qwen 2.5 14B | 65.3 | 34.6 | 78.4 | 50.5 | 83.9 | 82.4 | 70.6 | 81.1 | 79.3 | 21.1 | 70.8 |
| Open weights models 24-32B Parameters | | | | | | | | | | | |
| Gemma 2 27B | 61.3 | 49.0 | 72.7 | 67.5 | 80.7 | 63.2 | 35.1 | 70.7 | 75.9 | 33.9 | 64.6 |
| Mistral Small 24B | 67.5 | 43.2 | 80.1 | 78.5 | 87.2 | 77.3 | 65.9 | 83.7 | 66.5 | 24.4 | 68.1 |
| Qwen 2.5 32B | 68.1 | 39.1 | 82.3 | 48.3 | 87.5 | 82.4 | 77.9 | 84.7 | 82.4 | 26.1 | 70.6 |
| Gemma 3 27B | 71.3 | 63.4 | 83.7 | 69.2 | 91.1 | 83.4 | 76.2 | 81.8 | 69.1 | 30.9 | 63.9 |
| Fully-open models | | | | | | | | | | | |
| SmolLM2 1.7B | 34.2 | 5.8 | 39.8 | 30.9 | 45.3 | 51.6 | 20.3 | 34.3 | 52.4 | 16.4 | 45.3 |
| OLMo 7B 0424 | 33.1 | 8.5 | 34.4 | 47.9 | 23.2 | 39.2 | 5.2 | 48.9 | 49.3 | 18.9 | 55.2 |
| OLMO 2 7B | 56.5 | 29.1 | 51.4 | 60.5 | 85.1 | 72.3 | 32.5 | 61.3 | 93.3 | 23.2 | 56.5 |
| OLMO 2 13B | 63.5 | 39.5 | 63.0 | 71.5 | 87.4 | 82.6 | 39.2 | 68.5 | 89.7 | 28.8 | 64.3 |
| OLMO 2 32B | 68.8 | 42.8 | 70.6 | 78.0 | 87.6 | 85.6 | 49.7 | 77.3 | 85.9 | 37.5 | 73.2 |

Table 4: OLMO 2-INSTRUCT's performance vs closed and open-weights only models.

reasoning, and safety tasks from the same evaluation suite used by Lambert et al. (2024). Full descriptions of benchmark tasks in Appendix A.2.

Table 4 contains downstream results. We find **OLMO 2-INSTRUCT models are competitive with the best instruction-tuned open-weights models and even some popular proprietary models**. This shows the usefulness of OLMO 2 as a powerful base model that serves as an excellent starting point for fully open post-training research.

# 4 Pretraining Stability

In OLMO 2, we implemented and validated a number of techniques for mitigating training instability characterized by the presence of **sudden spikes** in the loss and gradient norm and **slow growth** in the magnitude of the gradient norm. We present the cumulative impact of these measures in Figure 2. We summarize the main interventions and their intuitions below, and for space, link to experimental results validating each choice.

**Removing repeated n-grams:** In prior experimental runs, we found a high prevalence of instances containing long, repeated n-gram sequences within training batches at which spikes occurred. An example of one such sequence is g40Dg40Dg40Dg40Dg40Dg40Dg40Dg40Dg40Dg40Dg40Dg40Dg40D... To mitigate, we filter our training data to remove documents with a sequence of 32 or more repeated n-grams, where an n-gram is any span of 1 to 13 tokens. We also implement an additional safeguard in the trainer that detects these sequences during data loading and masks them when computing the loss. We found this intervention results in a clear mitigation—though not complete elimination—of gradient spikes, and no effect on the slow growth in gradient norm. See further details and experimental results in Figure 4 in Appendix §F.1.

**Initialization:** Prior work used scaled initialization (Zhang et al., 2019; Gururangan et al., 2023; Ai2, 2024) that scaled input projections by $1/\sqrt{d_{\text{model}}}$, and output projections by $1/\sqrt{2 \cdot d_{\text{model}} \cdot \text{layer}_{\text{idx}}}$ at every layer. In other words, later layers were initialized to smaller values. In OLMO 2, we instead initialize all parameters with a mean of 0 and a

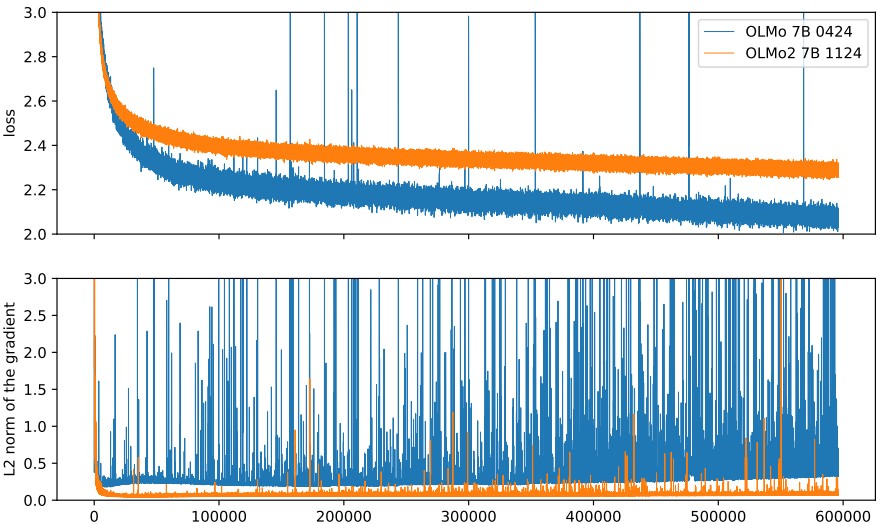

Figure 2: Training loss and gradient norm curves (over training steps) for OLMO 2 vs OLMO. The OLMO training run was marked by frequent loss spikes (top), often preceded by more frequent spikes in the gradient norm, which grew over time (bottom). We note that the training loss for OLMO 2 is higher because the underlying training data are different.

standard deviation of 0.02. We found empirically that this (1) better preserves the scale of activations and gradients across layers, allowing deep models to be trained more stably, and (2) this initialization transfers well across models of different widths. See further details and experimental results in Figure 5 in Appendix §F.2.

**Reordered norm and QK-norm** Following Liu et al. (2021) and Chameleon Team (2024), we apply layer normalization to the *outputs* of the MLP and attention blocks instead of the inputs. We also adopt QK-norm (Dehghani et al., 2023) which applies another normalization to the queries and keys in the attention block. Empirically, we found that, while neither of these changes yield stability improvements in isolation, together they improve both the growth and the spikiness of the L2 norm of the gradient. See further details and experimental results in Figure 8 in Appendix §F.3.1.

$\epsilon$ **in AdamW** In OLMO 2, we decreased the $\epsilon$ term in AdamW from 10E−5, a value commonly used in many LM training code bases (e.g. Megatron, OLMO), to 10E−8, the default value in PyTorch. We found the lower value allows for larger updates early in training, and helps the model learn faster during a period where we've typically seen more instability. As a result, the gradient norm settles much more quickly and remains permanently lower. See further details and experimental results in Figure 10 in Appendix §F.4.1.

**Weight decay on embeddings** A standard formulation of weight decay multiplies every parameter by $1 - (0.1 \cdot lr)$ at every step, a regularization term that discourages parameters from growing too large. We found in the case of token embeddings, it is too aggressive and results in very small embeddings. As discussed by Takase et al. (2024), small embeddings can produce large gradients in early layers because the Jacobian of layer_norm($x$) w.r.t. $x$ is inversely proportional to $\|x\|$, and, in early layers, the norm of the residual stream is essentially the norm of the embeddings. In OLMO 2, we experiment with the full range of remedies discussed in Takase et al. (2024), but found that they negatively impacted the speed of convergence. Instead, we simply turn off weight decay for embeddings and observe that embedding norms settle in a healthy region as training progresses. See further details and experimental results in Figure 11 in Appendix §F.4.2.

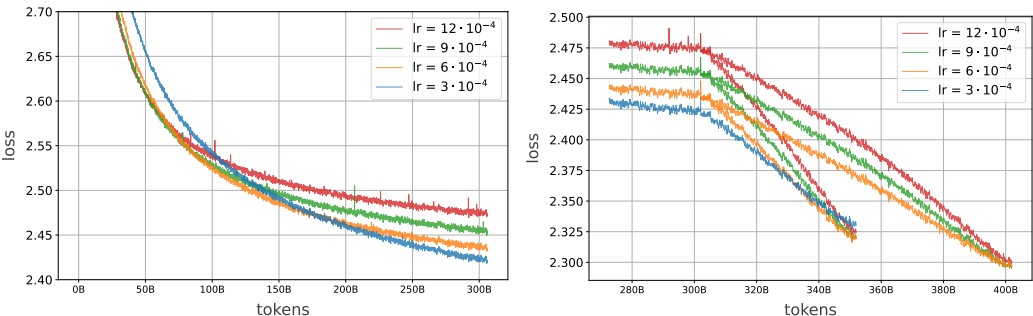

Figure 3: Higher learning rates perform better at first but are eventually overtaken by lower rates. However, linearly decaying the learning rate to zero over 50B or 100B tokens results in equivalent training loss.

# 5  Mid-training Recipe Development

As described in §2.1.2, our mid-training phase involves both linearly decaying learning rate to zero while changing the data curriculum. We discuss how we developed our final recipe.

## 5.1  Learning Rate Annealing

To determine how to set our learning rate schedule, including peak learning rate during Stage 1 (cosine decay) and training duration of Stage 2 (linear decay to zero), we trained identical 7B parameter models up to 300B tokens (Stage 1) with learning rates $6 \cdot 10E-4$, $9 \cdot 10E-4$, $12 \cdot 10E-4$, and $30 \cdot 10E-4$. Then, we annealed each model's learning rate linearly to zero over 50B or 100B tokens.

Figure 12 shows training loss under both stages. First, we found too high a learning rate ($30 \cdot 10E-4$) leads to training instability and loss spikes. Second, we found higher learning rates initially resulted in faster loss reduction, but lower learning rates eventually performed better. Yet, once we apply linear learning rate decay to zero, **the differences between the choice of learning rates largely disappear** as all paths result in similar final training loss, suggesting a **trade-off between pretraining and mid-training performance**. We find this result is consistent even when extending our experiments to 2T training tokens; see further details in Appendix §F.5. Overall, this contradicts common machine learning assumptions on the benefits of high learning rates (McCandlish et al., 2018), and aligns with observations from Wortsman et al. (2023) that training loss is not always improved by higher learning rates and smaller models' performance is largely invariant to learning rate when trained to the end of a cosine schedule.

## 5.2  Data Curriculum: DOLMINO MIX 1124

In this section, we describe our experimental process for curating our mid-training data.

**DOLMINO MIX 1124 High Quality Subset:** We first experiment with different data mixes by performing mid-training starting from an OLMO 2 7B checkpoint trained up to 4T tokens, switching to a candidate data mix (subsampled to 50B tokens). Candidate mixes were curated following these steps; exact mix specifications are in Table 13 in Appendix §G.1:

1. Start with knowledge-rich documents (e.g. scientific papers, books, Wikipedia, Stack Exchange), as seen in OLMO-0424 (Ai2, 2024),

2. Ablate different amounts of instruction data like Flan (Wei et al., 2021) (Ins) or math pretraining data like OpenWebMath (Paster et al., 2023) (Math),

3. Ablate different quality filters applied to DCLM, including choice of quality classifiers from DCLM (Li et al., 2024) (FT) or FineWeb (Penedo et al., 2024) (FW) and choice of threshold (e.g. $FT_7$ selects documents $\geq 0.7$, $FW_2$ selects documents $\geq$ "2" rating). This

| Mid-training mix | OLMES (MCF) | OLMES-Gen | MMLU (MCF) | GSM$^*$ |
|---|---|---|---|---|
| *n/a (pretrain checkpoint)* | 69.6 | 63.2 | 59.8 | 28.5 |
| PT Mix | 74.0 | 64.5 | 61.8 | 27.0 |
| Web $^{FT_7}$ | 73.5 | 64.1 | 61.9 | 24.5 |
| Web $^{FT_7}_{FW_3}$ | 73.5 | 63.0 | 62.4 | 30.5 |
| Web $^{FT_7}_{FW_2}$ | 75.2 | 63.8 | 63.1 | 28.5 |
| Web $^{FT_7}_{FW_2}$ + Ins | 74.2 | 64.1 | 63.0 | 46.0 |
| Web $^{FT_7}_{FW_2}$ + Math | **75.7** | 69.7 | 62.3 | **52.0** |
| Web $^{FT_7}_{FW_2}$ + Math + Ins | **75.7** | **70.2** | **63.1** | 46.5 |

Table 5: Ablation experiments for mid-training mixes (high quality subset); exact details on mixes in Appendix §G.1. Scores macro-averaged over OLMES benchmark tasks, grouped into multiple-choice (MCF), generative (Gen), MMLU, and our GSM$^*$ dev set.

always results in more data than needed for mid-training, so subsample until reach target token total (50B).

Table 5 presents ablation results. First, performing any mid-training without changing the pretraining data (PT Mix) improves on all tasks except math. Second, including higher quality web data further improves performance. Including instruction and math data yields the best performance. Our final choice for our high quality set combines all these elements.

**DOLMINO MIX 1124 Math Mix:** Even after mid-training on our best high-quality subset, OLMO 2 still showed weak math abilities. We address this with a specialized math mix focusing on instruction-based math problems rather than rely on general math pretraining corpora (e.g. OpenWebMath). Table 6 summarizes the mix. Our strategy is as follows:

- Use existing data TuluMath (Lambert et al., 2024) and GSM8K train (Cobbe et al., 2021),

- Filter synthetic textbooks from open repos[4,5] and M-A-P Matrix (Zhang et al., 2024a) using a fastText classifier we distilled from 10k GPT-4o predicted labels categorizing OpenWebMath documents as math/non-math; this filtering procedure follows from Math-Coder2 (Lu et al., 2024).

- Filter existing data Metamath (Yu et al., 2023) and CodeSearchNet (Husain et al., 2019) using our same classifier,

- Create TinyGSM-MIND, 6.5B tokens of synthetic math data from rewritten versions of Tiny-GSM (Liu et al., 2023a), a collection of 11M synthetic GSM8K-like questions and Python code answers. To do this, we (1) filter to QA pairs including answers with executable code and only include variable assignment statements, (2) annotate each line of code with an assignment operator with the numerical value of the resulting variable, and (3) use Qwen2.5-7B-Instruct to rewrite annotated examples in the style of MIND (Akter et al., 2024) using the 'Two Students' and 'Problem Solving' prompts.

- Create DOLMINOSYNTHMATH, 28M synthetic math tokens for solving raw math calculations and simple math problems. It's comprised of three subsets: (1) 11M generated tokens of basic math QA pairs (e.g., "77 * 14 = 1078") and diverse natural language prompts. (2) 7,924 examples synthetically perturbing numbers in GSM8K training examples using a custom computational graph parser. (3) MIND-rewriting (Akter et al., 2024) of each of the GSM8K training examples using Qwen2.5-7B-Instruct (Qwen et al., 2024).

**Microannealing:** Experimentation with many small data sources, especially while also iterating on synthetic pipeline parameters (e.g. prompts, filters), requires a reliable experimental procedure for rapid decision-making. We develop a procedure called *microannealing*

---

[4] 🗄 datasets/ajibawa-2023/Maths-College

[5] 🗄 datasets/ajibawa-2023/Education-College-Students

| Source | Type | Tokens | Words | Bytes | Docs |
|---|---|---|---|---|---|
| **High Quality Subset** | | | | | |
| DCLM-Baseline | High quality web | 752B | 670B | 4.56T | 606M |
| FLAN | Instruction data | 17.0B | 14.4B | 98.2B | 57.3M |
| peS2o | Academic papers | 58.6B | 51.1B | 413B | 38.8M |
| Wikipedia & Wikibooks | Encyclopedic | 3.7B | 3.16B | 16.2B | 6.17M |
| Stack Exchange | Q&A | 1.26B | 1.14B | 7.72B | 2.48M |
| **Subtotal** | | **832.6B** | **739.8B** | **5.09T** | **710.8M** |
| **Math Mix** | | | | | |
| TuluMath | Synthetic math | 230M | 222M | 1.03B | 220K |
| GSM8K Train | Math | 2.74M | 3.00M | 25.3M | 17.6K |
| Filtered Synth Books | Synthetic Math | 3.87B | 3.71B | 18.4B | 2.83M |
| Filtered Metamath | Math | 84.2M | 76.6M | 741M | 383K |
| Filtered CodeSearchNet | Code | 1.78M | 1.41M | 29.8M | 7.27K |
| TinyGSM-MIND | Synthetic math | 6.48B | 5.68B | 25.52B | 17M |
| DOLMINOSYNTHMATH | Synthetic math | 28.7M | 35.1M | 163M | 725K |
| **Subtotal** | | **10.7B** | **9.73B** | **45.9B** | **21.37M** |

Table 6: Composition of OLMO 2 mid-training data (DOLMINO MIX 1124).

that performs mid-training on 100% of a candidate dataset, validating whether the candidate data source provides nonzero performance improvement on our math dev set GSM*. Similar procedures have been seen in Blakeney et al. (2024) and Grattafiori et al. (2024), but we differ in two key ways: (1) using *only* the candidate data instead of more commonly-seen 70/30 or 50/50 mixes with web data and (2) not restricting to a target training length (e.g. 50B tokens). This maximizes chances for us to see signal in even small yet impactful datasets.[6]

## Conclusion

We introduce OLMO 2 and OLMO 2-INSTRUCT, a family of fully open 7B, 13B and 32B parameter language models trained on up to 6T tokens. Both the base and instruct models are competitive with other open-weight models in their size categories such as Qwen 2.5, Gemma 2, and Llama 3.1. We detail the substantial contributions required to build competitive language models including architecture improvements for stability and innovations in late-stage training data. We release all training and evaluation code, datasets, checkpoints, and logs required to reproduce and expand on the models. OLMO 2 marks continued progress in open-source language models, building an ecosystem for research, one where new training methods and techniques can be understood and shared.

## Author Contributions

A successful team project like OLMO would not be possible without the fluid contributions of many teammates across formal team boundaries. As not all of these can be captured, we indicate each authors' primary contributing role in OLMO 2. Authors are listed in alphabetical order:

- For base model development, including training and data curation: Shane Arora, Akshita Bhagia, Christopher Clark, Allyson Ettinger, Dirk Groeneveld, Yuling Gu, David Heine-

---

[6]For example, mid-training from a 7B checkpoint at 4T tokens on original TinyGSM data actually *degraded* performance on GSM*, which motivated us to explore MIND-style rewriting.

man, Matt Jordan, Jiacheng Liu, Kyle Lo, William Merrill, Tyler Murray, Jake Poznanski, Dustin Schwenck, Luca Soldaini, Oyvind Tafjord, David Wadden, and Pete Walsh.

- For instruct model development, including training and data curation: Faeze Brahman, Pradeep Dasigi, Nouha Dziri, Yuling Gu, Shengyi Huang, Hamish Ivison, Nathan Lambert, Saumya Malik, Lester James V. Miranda, Jacob Morrison, Valentina Pyatkin, Oyvind Tafjord, and Christopher Wilhelm.
- For operational support, including program management, legal guidance, release process, and more: Taira Anderson, David Atkinson, Crystal Nam, and Aman Rangapur.
- For Ai2 cluster setup and support: Michal Guerquin, Michael Schmitz, Sam Skjonsberg, and Michael Wilson
- For mentorship and advising: Ali Farhadi, Hannaneh Hajishirzi, Pang Wei Koh, Noah A. Smith, and Luke Zettlemoyer.

Authorship for this work was determined by those making direct contributions to the OLMo 2 models, related artifacts, and their release. Core contributors are recognized for their sustained, significant contributions critical to the success of the OLMo 2 project.

## Acknowledgments

This work would not be possible without the support of our colleagues at Ai2:

- We thank Ben Bogin, Tim Dettmers, Ananya Harsh Jha, Ani Kembhavi, Matt Deitke, Ian Magnusson, Sewon Min, Niklas Muennighoff, Yizhong Wang, Alexander Wettig, and Valentin Hofmann for helpful research discussions and sharing of relevant findings across related projects.
- We thank Taylor Blanton, Byron Bischoff, Yen-Sung Chen, Arnavi Chheda, Jesse Dodge, Karen Farley, Huy Tran, Eric Marsh, Chris Newell, and Aaron Sarnat for building the Ai2 Playground for model demos.
- We thank Yoganand Chandrasekhar, Johann Dahm, Fangzhou Hu, and Caroline Wu for their work on the Ai2 cluster.
- We also thank others at Ai2 for many indirect contributions to the project: Robert Berry, Alex Buraczynski, Jennifer Dumas, Jason Dunkelberger, Rob Evans, David Graham, Regan Huff, Jenna James, Rodney Kinney, Bailey Kuehl, Sophie Lebrecht, Jaron Lochner, Carissa Schoenick, Will Smith, Sruthi Sreeram, Brooke Vlahos, Alice Wang, Caitlin Wittlif, Jiangjiang Yang.

We also appreciate conversations with and feedback from Cody Blakeney, Mansheej Paul, Jonathan Frankle, Armen Aghajanyan, Akshat Shrivastava, Mike Lewis, and John Schulman.

OLMo 2 would not have been possible without the support of many other institutions. In particular, we thank Google for their support in setting up the training environment for OLMo 2 and Cirrascale for their on-going support of Ai2's cluster. We also acknowledge the National Artificial Intelligence Research Resource (NAIRR) Pilot and Microsoft Azure for providing inference credits in support of this project.

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

# A  Evaluation Framework

We evaluate OLMO 2 using OLMES (Gu et al., 2024), a unified, standardized evaluation suite and toolkit to guide the development and assess performance of language models.

## A.1  Base Model Eval

The OLMO 2 base models are evaluated on 20 tasks, consisting of 10 multiple-choice tasks, 5 generative tasks, and 5 additional held-out tasks not utilized during model development. See Table 7 for the list of tasks along with details of the task formulations following the principles of the OLMES standard, described further below. For this version of the paper, due to space constraints, we report results for only 10 of these tasks; for full results see OLMo et al. (2024).

| task | split | # inst (total) | # shots | metric | reference |
|---|---|---|---|---|---|
| *Multiple-choice tasks* | | | | | |
| ARC-Challenge (ARC$_C$) | Test | 1172 | 5 | pmi | (Clark et al., 2018) |
| ARC-Easy | Test | 1000 (2376) | 5 | char | (Clark et al., 2018) |
| BoolQ | Val | 1000 (3270) | 5 | none | (Clark et al., 2019) |
| CommonsenseQA | Val | 1221 | 5 | pmi | (Talmor et al., 2019) |
| HellaSwag (HS) | Val | 1000 (10042) | 5 | char | (Zellers et al., 2019) |
| MMLU[†] | Test | 14042 | 5 | char | (Hendrycks et al., 2021a) |
| OpenbookQA | Test | 500 | 5 | pmi | (Mihaylov et al., 2018) |
| PIQA | Val | 1000 (1838) | 5 | char | (Bisk et al., 2020) |
| Social IQa | Val | 1000 (1954) | 5 | char | (Sap et al., 2019) |
| WinoGrande (WG) | Val | 1267 | 5 | none | (Sakaguchi et al., 2020) |
| *Generative tasks* | | | | | |
| CoQA | Val | 7983 | 0 | F1 | (Reddy et al., 2019) |
| DROP | Val | 1000 (9536) | 5 | F1 | (Dua et al., 2019) |
| Jeopardy | Test | 2117 | 5 | F1 | (MosaicML, 2024) |
| Natural Questions (NQ) | Val | 1000 (3610) | 5 | F1 | (Kwiatkowski et al., 2019) |
| SQuAD | Val | 1000 (10570) | 5 | F1 | (Rajpurkar et al., 2016) |
| *Held-out tasks* | | | | | |
| AGIEval English (AGI) | Test | 2646 | 1 | MCF | (Zhong et al., 2024) |
| BBH | Test | 6511 | 3 (CoT) | EM | (Suzgun et al., 2022) |
| GSM8K (GSM) | Test | 1319 | 8 (CoT) | EM | (Cobbe et al., 2021) |
| MMLU-Pro (MMLU$_P$) | Test | 12032 | 5 | MCF | (Wang et al., 2024) |
| TriviaQA (TQA) | Val | 7993 | 5 | F1 | (Joshi et al., 2017) |

Table 7: Details of OLMES benchmarks used to evalute OLMO 2, with standardized choices of dataset split, number of instances to use, along with total number if sampling was used. For multiple-choice tasks, when using the Cloze/Completion Formulation (CF), the "metric" column specifies which normalization scheme to use. Following the OLMES standard, we evaluate each model using both the MCF (Multiple-Choice Formulation) and CF formulations, and the best performing one is used. For efficiency reasons, we limit MMLU and held-out multiple-choice evaluations to MCF only as all the relevant models strongly prefer that format for these tasks.

**Multiple-choice tasks**  We use the formulation of the 10 multiple-choice tasks defined in the OLMES evaluation standard (Gu et al., 2024). OLMES (Open Language Model Evaluation Standard) is a set of principles and associated standard (with a reference implementation in the OLMES system framework) for reproducible LM evaluations that is open, practical, and documented, providing recommendations guided by experiments and results from the literature (Biderman et al., 2024; Gao et al., 2023). For multiple-choice tasks it is designed

to support comparisons between smaller base models that require the cloze/completion formulation of multiple-choice questions (score each answer completion separately) against larger models that can utilize the multiple-choice formulation. To make our evaluations reproducible, we follow the OLMES standard in prompt formatting, choice of in-context examples, probability normalization, and all other details. We report the exact evaluation splits and numbers of instances in Table 7.

**Generative tasks** Following the principles of OLMES (Gu et al., 2024), such as prompt formatting and having 5-shot curated in-context examples, we also evaluated on a suite of generative tasks, OLMES-Gen. This suite covers factual knowledge tasks (Natural Questions (Kwiatkowski et al., 2019) and Jeopardy (MosaicML, 2024)) and tasks testing reading comprehension (SQuAD (Rajpurkar et al., 2016), DROP (Dua et al., 2019), and CoQA (Reddy et al., 2019)). For CoQA, the task comprises presenting a passage followed by a conversation so far, where each turn in the conversation contains a question and an answer. In this case, the previous question and answer pairs serve to guide the model in terms of the output format, and we do not include additional few-shot examples. For all other tasks, we follow OLMES in using 5-shot curated in-context examples. As the list of gold answers for these tasks are often incomplete, we use F1 as the primary metric to give partial credit when models produce answers that partially match. The task details of OLMES-Gen are summarized in Table 7.

**Held-out tasks** We also evaluate on a held-out suite of tasks that were not used when making decisions during model development. This suite includes advanced admission and qualification exams (AGIEval English[7] (Zhong et al., 2024)), tasks believed to be challenging to LMs (BigBenchHard, BBH; Suzgun et al., 2022), math reasoning (GSM8K; Cobbe et al., 2021), a more challenging and reasoning-focused extension of MMLU (MMLU Pro; Wang et al., 2024), and an unseen factual knowledge task (TriviaQA; Joshi et al., 2017). We use existing in-context examples where available - for GSM8K, we use the 8-shot CoT examples from Wei et al. (2022); for BBH we use the 3-shot CoT prompts from the original dataset; in evaluating MMLU-Pro, we used 5-shot examples from the original dataset. We use a 1-shot (with passage context, no CoT) prompt for AGIEval English, and a manually curated 5-shot examples from the train set for TriviaQA. Note that for the case of GSM8K, we never evaluated our models on the entire test set during the development stage, instead we use 200 examples to inform choices during development (e.g., choices of annealing mixtures); in Section 5 we refer to this 200-example subset as GSM*.

### A.2 Instruct Model Eval

**Instruct tasks** We perform instruct model evaluation based on existing practices in current literature using the OLMES benchmark suite (Gu et al., 2024) using the configuration reported in Lambert et al. (2024).

See Table 8 for a list of instruct tasks along with their configurations. These tasks include chat variations of our held-out tasks (GSM8k; Cobbe et al., 2021, BBH; Suzgun et al., 2022), additional long-tail knowledge (PopQA; Mallen et al., 2022), misconception (TruthfulQA; Lin et al., 2021) and instruction-following tasks (IFEval; Zhou et al., 2023, AlpacaEval 2; Dubois et al., 2024). For our MMLU instruct evaluation, we use the CoT version from Lambert et al. (2024) using their prompt asking the model to "summarize" its reasoning before answering the question. We evaluate Python code completion (HumanEval; Chen et al., 2021, HumanEval+; Liu et al., 2023b) and competition MATH (Hendrycks et al., 2021b) with the same setup and answer extraction in OLMES.

---

[7]Specifically these 8 tasks: aqua-rat, logiqa-en, lsat-ar, lsat-lr, lsat-rc, sat-en, sat-math, gaokao-english

| Category | Task | CoT | # shots | Chat | Multiturn ICL | Metric |
|---|---|---|---|---|---|---|
| | | *Instruct tasks* | | | | |
| Knowledge Recall | MMLU | ✓ | 0 | ✓ | ✗ | EM |
| | PopQA | ✗ | 15 | ✓ | ✓ | EM |
| | TruthfulQA | ✗ | 6 | ✓ | ✗ | MC2 |
| Reasoning | BigBenchHard | ✓ | 3 | ✓ | ✓ | EM |
| | DROP | ✗ | 3 | ✗ | N/A | F1 |
| Math | GSM8K | ✓ | 8 | ✓ | ✓ | EM |
| | MATH | ✓ | 4 | ✓ | ✓ | Flex EM |
| Coding | HumanEval | ✗ | 0 | ✓ | N/A | Pass@10 |
| | HumanEval+ | ✗ | 0 | ✓ | N/A | Pass@10 |
| Instruction Following | IFEval | ✗ | 0 | ✓ | N/A | Pass@1 (prompt; loose) |
| | AlpacaEval 2 | ✗ | 0 | ✓ | N/A | LC Winrate |
| Safety | Tülu 3 Safety | ✗ | 0 | ✓ | N/A | Average[*] |

Table 8: Details of OLMES benchmarks used for to evaluate OLMO 2-INSTRUCT. **CoT** are evaluations run with chain of thought prompting (Wei et al., 2022). **#Shots** is the number of in-context examples in the evaluation template. **Chat** refers to whether we use a chat template while prompting the model. **Multiturn ICL** refers to a setting where we present each in-context example as a separate turn in a conversation (applicable only when a chat template is used and # Shots is not 0). [*]Average over multiple sub-evaluations

## B OLMO 2 1B

While the goal of this work is to develop development recipes for our target 7B, 13B and 32B sizes, often it is useful to perform experimentation at the 1B model size. We define OLMO 2 1B similar to OLMO 2 7B, but with the following departures:

- **Layers:** 16 instead of 32
- **Hidden Size ($d_{model}$:** 2048 instead of 4096
- **Attention Heads (Q/KV):** 16/16 (MHA) instead of 32/32 (MHA)
- **Batch Size:** 512 instead of 1024
- **Peak LR:** $4.0 \cdot 10E{-}4$ instead of $3.0 \cdot 10E{-}4$

## C OLMO 2 Tokenizer

| Tokenizer | OLMES (CF) | OLMES Gen | MMLU (CF) |
|---|---|---|---|
| OLMO 1 tokenizer | 59.8 | 42.4 | 34.8 |
| OLMO 2 tokenizer | 60.6 | 42.7 | 35.2 |

Table 9: Comparison of OLMO 1 and OLMO 2 tokenizers on a 1B model pretrained for 100B tokens from DCLM baseline. Following Gu et al. (2024), OLMES and MMLU use CF format, which is more informative for small models.

Comparing OLMO 1 and OLMO 2 tokenizers at a smaller scale in Table 9, we notice measurable gains even in downstream tasks, even not accounting for the disadvantage imposed on the larger tokenizer given this smaller model experiment (Tao et al. (2024)).

This result gave us confidence to proceed with the switch to the new tokenizer as we expect improvement coming from larger vocabulary to be more decisive at larger scales and for models trained on more tokens.

# D    OLMO 2 Stage 1 Data: Pretraining

| Source | Type | Tokens | Words | Bytes | Docs |
|---|---|---|---|---|---|
| **Pretraining** | | | | | |
| DCLM-Baseline | Web pages | 3.71T | 3.32T | 21.32T | 2.95B |
| StarCoder | Code | 83.0B | 70.0B | 459B | 78.7M |
| peS2o | Academic papers | 58.6B | 51.1B | 413B | 38.8M |
| arXiv | STEM papers | 20.8B | 19.3B | 77.2B | 3.95M |
| OpenWebMath | Math web pages | 12.2B | 11.1B | 47.2B | 2.89M |
| Algebraic Stack | Math proofs code | 11.8B | 10.8B | 44.0B | 2.83M |
| Wikipedia & Wikibooks | Encyclopedic | 3.7B | 3.16B | 16.2B | 6.17M |
| **Total** | | **3.90T** | **3.48T** | **22.38T** | **3.08B** |

Table 10: **Composition of the pretraining data for OLMO 2**. The OLMO 2 MIX 1124 is composed of StarCoder (Li et al., 2023; Kocetkov et al., 2022), peS2o (Soldaini & Lo, 2023), web text from DCLM (Li et al., 2024) and Wiki come from Dolma 1.7 (Soldaini et al., 2024). arXiv comes from Red-Pajama (Together AI, 2023), while OpenWebMath (Paster et al., 2023) and Algebraic Stack come from ProofPile II (Azerbayev et al., 2023).

We adopt the pretraining data mix in OLMOE (Muennighoff et al., 2024). This section describes the content of this data in further detail. See Table 10 for exact counts.

- From DCLM, we use the "*baseline 1.0*" mix.[8]
- From Dolma 1.7 (Soldaini et al., 2024), we use the arXiv (Together AI, 2023), OpenWeb-Math (Paster et al., 2023), Algebraic Stack, peS2o (Soldaini & Lo, 2023), and Wikipedia subsets. arXiv, OpenWebMath, and Algebraic Stack were originally part of ProofPile II (Azerbayev et al., 2023).
- From StarCoder (Li et al., 2023), we use permissively-licensed repositories from GitHub (Kocetkov et al., 2022) with any document from a repository with fewer than 2 stars on GitHub removed.

Through manual inspection of the StarCoder mix from OLMOE, we discovered numerous documents encoded in binary format or containing mostly numerical content. Thus, we perform an additional round of heuristic filtering to remove this low quality items, discarding documents whose most frequent word constitutes over 30% of the document, or whose top-2 most frequent words constitute over 50% of the document.

# E    OLMO 2 Model Architecture

In OLMO (Groeneveld et al., 2024), we modified the decoder-only transformer architecture (Vaswani et al., 2017) with:

- **No biases:** We exclude all bias terms following (Groeneveld et al., 2024; Chowdhery et al., 2022, *inter alia*).
- **SwiGLU activation function:** We use the SwiGLU activation function (Shazeer, 2020) and set the corresponding hidden size to approximately $\frac{8}{3}d$, but increased to the closest multiple of 128 (11, 008 for our 7B model) to improve throughput.

---

[8]Available at ⬛ mlfoundations/dclm-baseline-1.0

- **Rotary positional embeddings (RoPE):** We replace absolute positional embeddings with rotary positional embeddings (RoPE; Su et al., 2021).

In OLMO-0424 (Ai2, 2024), we made further modifications for training stability and downstream performance:

- **QKV Clipping:** For training stability, also as seen in DBRX (Databricks, 2024).
- **Increased context:** From 2048 to 4096.

For OLMO 2, we start from a similar architecture but then depart from these prior architectures with modifications:

- **RMSNorm:** We use the RMSNorm (Zhang & Sennrich, 2019) variant of LayerNorm (Ba et al., 2016) without a bias term to normalize activations, instead of nonparametric LayerNorm.
- **Reordered norm:** We normalize the outputs to the attention and feedforward (MLP) layers within each transformer block, instead of the inputs. So the formula for each block becomes:

$$h := x + \text{RMSNorm}(\text{Attention}(x)) \tag{1}$$
$$h_{\text{out}} := h + \text{RMSNorm}(\text{MLP}(x)) \tag{2}$$

where $x$ is the input to the layer, $h$ is an intermediate hidden state, and $h_{\text{out}}$ is the output. This strategy was first proposed by Liu et al. (2021) to stabilize training.

- **QK-norm:** Following Dehghani et al. (2023) we normalize the key and query projections with RMSNorm before calculating attention. This avoids attention logits being too large, which can lead to training loss divergence.
- **Z-Loss:** Following Chowdhery et al. (2022), Chameleon Team (2024), and Wortsman et al. (2023), we adopt z-loss regularization, as it has been empirically shown to improve run stability.
- **RoPE value:** We increase the RoPE $\theta$ to 500,000 from 10,000. This approach increases the resolution of positional encoding, matching Grattafiori et al. (2024).

| | OLMO 1 (0224) | OLMO-0424 | OLMO 2 |
|---|---|---|---|
| **Biases** | None | None | None |
| **Activation** | SwiGLU | SwiGLU | SwiGLU |
| **RoPE** $\theta$ | $1 \cdot 10E4$ | $1 \cdot 10E4$ | $5 \cdot 10E5$ |
| **QKV Normalization** | None | Clip to 8 | QK-Norm |
| **Layer Norm** | non-parametric | non-parametric | RMSNorm |
| **Layer Norm Applied to** | Inputs | Inputs | Outputs |
| **Z-Loss Weight** | 0 | 0 | 10E−5 |
| **Weight Decay on Embeddings** | Yes | Yes | No |

Table 11: Summary of how OLMO 2's model architecture differs from OLMO.

## F   Pretraining Stability

### F.1   Repeated n-Grams

Figure 4 shows the effect of masking the loss of input sequences containing repeated n-grams on the gradient norm, demonstrating that broad removal of such sequences across training decreases the frequency of spikes, on average.

While we have found these sequences are often associated with spikes, we note that this relationship is not deterministic:

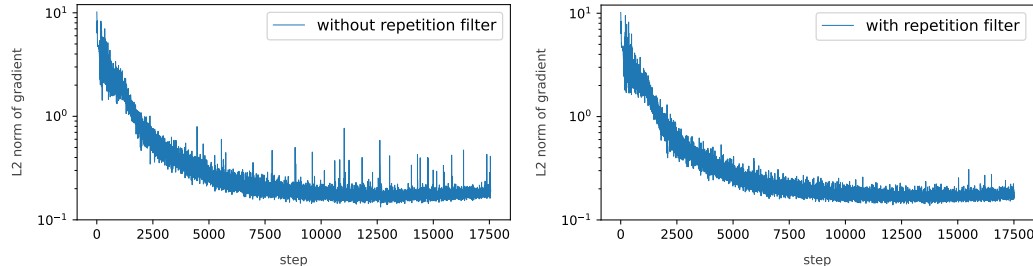

Figure 4: Comparison of the gradient norm for two runs, one without n-gram filter, and one with. Ignoring long repetitive sequences of n-grams eliminates many spikes.

- The same n-gram sequence may spike for a larger model but not for a smaller model trained on the same data.
- The same n-gram sequence may spike for one data training ordering, but not after the data is reshuffled.
- The same n-gram sequence associated with a spike can also be found elsewhere in training batches that did not spike.

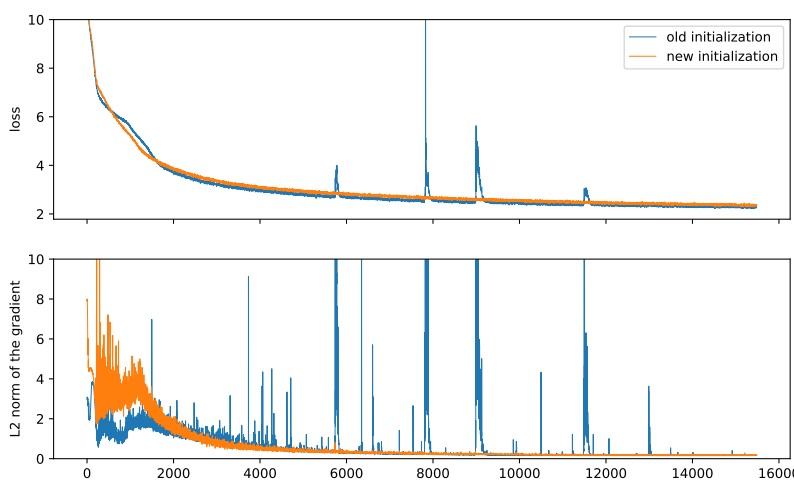

Figure 5: The older initialization shows instabilities quickly, while OLMO 2 stays stable.

### F.2 Model Initialization

Figure 5 shows improved training stability from OLMO 2's initialization scheme. In OLMO 2, we initialize every parameter from a normal distribution with a mean of 0 and a standard deviation of 0.02.

We perform several analyses to study the impact of initialization, showing that OLMO 2's initialization is superior to scaled initialization. Our empirical analysis suggests it better preserves the scale of activations and gradients across layers, allowing deep models to be trained more stably, and it exhibits properties associated with hyperparameter transfer across models of different widths. These two properties together give us confidence that deep models will train stably and that the initialization hyperparameters of our smaller models could transfer to larger scales.

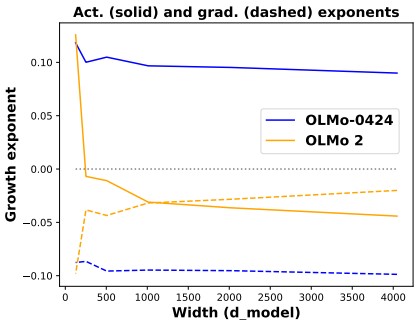

Figure 6: Across widths, growth exponents for the OLMO 2 initialization are closer to 0 compared to the OLMO-0424 initialization, which suggests deeper models will train more stably.

**Gradient and activation growth** A fundamental concern for training deep networks is ensuring that the activations and gradients do not blow up or vanish across layers, causing learning to become unstable or stagnate. Rather, we want the scale of the activations and gradients to remain roughly the same from layer to layer. Inspired by recent related work (Cowsik et al., 2024), we evaluate different candidate initializations in terms of how they affect the 2-norm of the activations and gradients across layers. Concretely, we randomly initialize a model, pass 50 random documents from The Pile (Gao et al., 2021) through it, and collect the activations and gradients (of loss with respect to the activations) at the initial and final layers (ignoring embeddings). We then average these tensors across documents and time steps to get vectors $\boldsymbol{v}$ at the initial layer and $\boldsymbol{v'}$ at the final layer, both of length $d_{\mathrm{model}}$. Finally, we compute the following measure of expansion or contraction across layers, which we call the *growth exponent*:

$$\lambda = \frac{1}{n_{\mathrm{layers}}} \log\left( \frac{\|\boldsymbol{v'}\|}{\|\boldsymbol{v}\|} \right)$$

We compute $\lambda$ for both the activations and gradients. Ideally, both $\lambda$'s remain near 0, indicating that the activations and gradients do not explode or vanish across layers. Figure 6 plots the growth exponents for different randomly initialized models as a function of their widths (4096 corresponds to a full 7B model). Crucially, the growth exponent for OLMO 2 is closer to 0 than for OLMO-0424 across model widths. This suggests the OLMO 2 initialization will be more stable when training deep models in low precision, as both the activations and the gradients are more resistant to exploding or vanishing across layers compared to the OLMO-0424 initialization.

**Hyperparameter transfer across width** Another appealing property of the new initialization is that it scales the activation and gradient norms with width ($d_{\mathrm{model}}$) in a way that has been argued theoretically to be important for hyperparameter transfer across different widths. Specifically, Yang et al. (2024b) suggest that a sufficient condition for hyperparameter transfer across width is that the magnitude of each activation scalar value and its update (learning rate times gradient) remain fixed as width increases. Equivalently, the norms of the activations and their update vectors should positively correlate with $\sqrt{d_{\mathrm{model}}}$. We plot the activation and gradient norms at initialization against $\sqrt{d_{\mathrm{model}}}$ in Figure 7. Crucially, the gradient norm is more positively correlated with $\sqrt{d_{\mathrm{model}}}$ for OLMO 2 compared to OLMO-0424. Combined with Yang et al. (2024b), this suggests that, with an initial learning rate independent of model width, the new OLMO 2 initialization will transfer better across different model widths compared to the OLMO-0424 initialization.

**Spike score** Since fast spikes are difficult to understand with contemporary graphing tools, we compute a *spike score* as an objective measure. Concretely, We define the spike score as the percentage of values in a time series that are at least seven standard deviations away

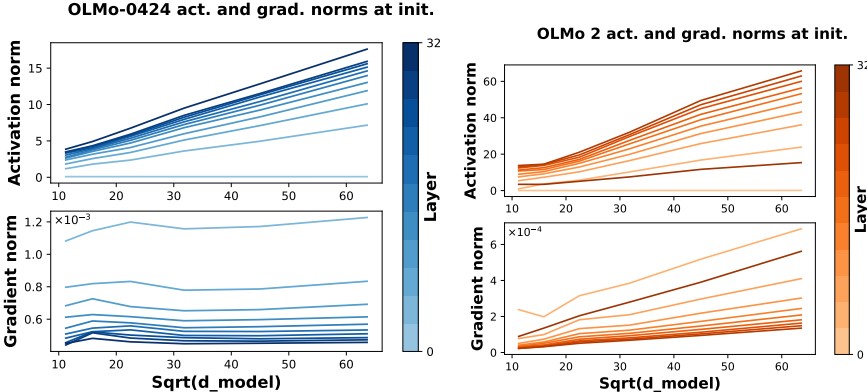

Figure 7: Activation and gradient norms vs. $\sqrt{d_{\text{model}}}$ for the OLMO-0424 and OLMO 2 initializations. Crucially, the gradient norms for OLMO 2 positively correlate with $\sqrt{d_{\text{model}}}$, which they did not for the OLMO-0424 initialization. This suggests the OLMO 2 initialization will show better hyperparameter transfer across widths (Yang et al., 2024b).

from a rolling average of the last $1,000$ values[9]. We use spike score primarily on training loss and L2 norm of the gradient, but the measure can be computed on any time series.

**Empirical results** To experiment with model initialization, we first create a baseline rune that reproduces spikes quickly. We do so by mainly reducing the warmup period. The effect was immediate and dramatic (Figure 5), and persists across model scales and token counts. In our ablation, the new initialization had no loss spikes, and the spike score for the L2 norm of the gradient went from 0.40 to 0.03. The new initialization converges slightly slower; we make up for this difference by improving other hyperparameter settings (Section §F.4).

### F.3 Architecture Improvements

#### F.3.1 Reordered norm and QK-norm

Figure 8 shows the effect of applying the layer normalization to the *outputs* of the MLP and attention blocks instead of the inputs. We further apply another normalization, also RMSNorm, to the queries and keys in the attention block. In isolation, neither of these changes yield good results, but together they improve both the growth and the spikiness of the L2 norm of the gradient. The following table summarizes the difference in the location of the layer normalization:

| OLMO-0424 | OLMO 2 |
|---|---|
| $h := x + \text{Attention}(\text{LN}(x))$ | $h := x + \text{RMSNorm}(\text{Attention}(x))$ |
| $h_{\text{out}} := h + \text{MLP}(\text{LN}(h))$ | $h_{\text{out}} := h + \text{RMSNorm}(\text{MLP}(h))$ |

$x$ is the input to the layer, $h$ is an intermediate hidden state, and $h_{\text{out}}$ is the output.

Liu et al. (2021) first introduced layer norm the idea of reordering layer norm. It was subsequently picked up by Chameleon Team (2024). QK-norm was first developed in Dehghani et al. (2023).

#### F.3.2 Z-Loss

Following Chowdhery et al. (2022), Chameleon Team (2024), and Wortsman et al. (2023), we apply z-loss regularization by adding $10\text{E}{-4} \cdot \log^2 Z$ to our loss function, where $Z$ is the

---

[9]Spike score is conceptually similar to spike mitigation proposed by Karpathy (2024).

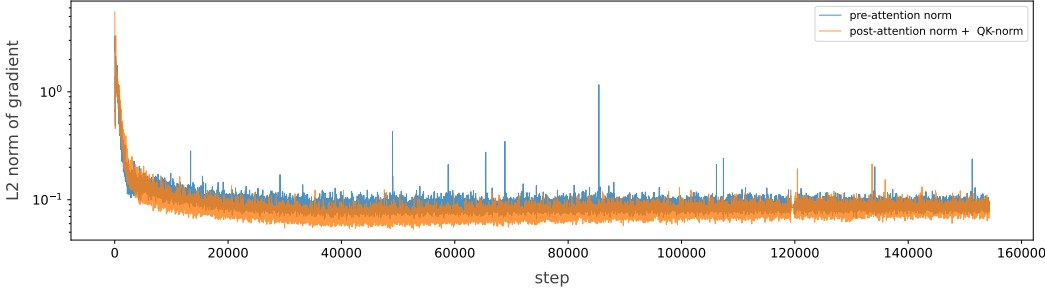

Figure 8: Applying layer norm after the attention and feedforward layers along with a QK-norm improves stability compared to a more standard pre-attention layer norm. These changes reduce the spike score of the gradients from 0.108 to 0.069 when applied together.

denominator in the softmax over the logits. This discourages the activations in the final softmax from growing too large, improving the stability of the model.

Figure 9 shows a stark difference between the z-loss implementation of the popular Flash Attention library (Dao, 2024), and an implementation using only Python primitives. Apart from the attention mechanism it is known for, Flash Attention also provides an optimized implementation of cross-entropy loss, which includes a version of z-loss. To retain flexibility in settings that are not compatible with Flash Attention, we have a separate implementation written in PyTorch. Both implementations produce the same result in the forward pass, but exhibit different behavior in the backward pass. We suspect the root cause lies in differences in precision. In our experiments, this does not affect cross entropy loss during training, or the model's performance on downstream tasks. However, out of an abundance of caution we abandon the fork with custom z-loss implementation and re-train from the original point of divergence. During a training run we cannot switch implementations safely, so we avoid doing so as much as possible.

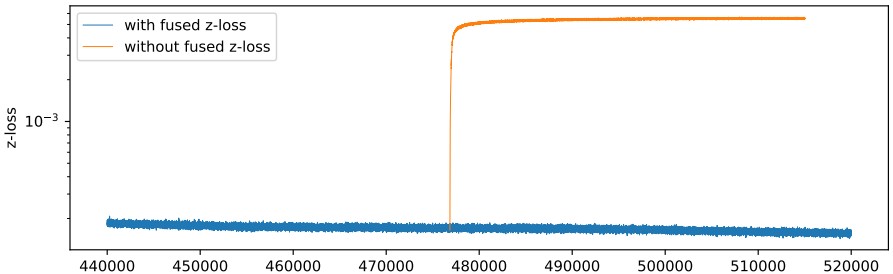

Figure 9: Flash Attention's implementation of z-loss does not match a manual implementation in PyTorch. While the forward pass produces the same number, differences in the backwards pass cause the curves to diverge.

### F.4   Hyperparameter Improvements

#### F.4.1   $\epsilon$ in AdamW

Figure 10 shows the result of decreasing the AdamW $\epsilon$ from 10E−5 to 10E−8. 10E−8 is the default in PyTorch, but some popular LM training code bases come with a default of 10E−5. The lower value allows for larger updates early in training, and helps the model learn faster during a period where we've typically seen a lot of instability. As a result, the gradient norm settles much more quickly and remains permanently lower.

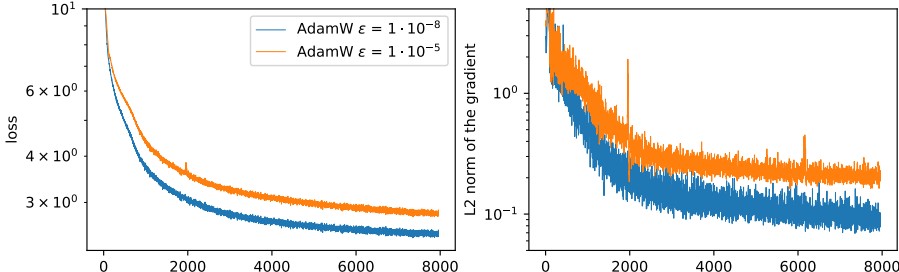

Figure 10: Setting AdamW's $\epsilon$ to 10E−8 lowers and stabilizes the norm of the gradient early in training. The training loss also improves faster. This trend continues even with runs that are longer than what is shown here.

### F.4.2   *Weight decay on embeddings*

Figure 11 shows the change in training dynamics following a decision to exclude weight decay for embeddings. OLMO uses a standard formulation of weight decay, where every parameter is multiplied by $1 - (0.1 \cdot lr)$ at every step. This regularization term discourages parameters from growing too large, but in the case of token embeddings it overshoots the mark and results in very small embeddings. As discussed by Takase et al. (2024), small embeddings can produce large gradients in early layers because the Jacobian of layer_norm($x$) w.r.t. $x$ is inversely proportional to $\|x\|$, and, in early layers, the norm of the residual stream is essentially the norm of the embeddings. We experiment with the full range of remedies discussed in Takase et al. (2024), but found that they impacted the speed of convergence. Instead, we simply turn off weight decay for embeddings and observe that embedding norms settle in a healthy region as training progresses.

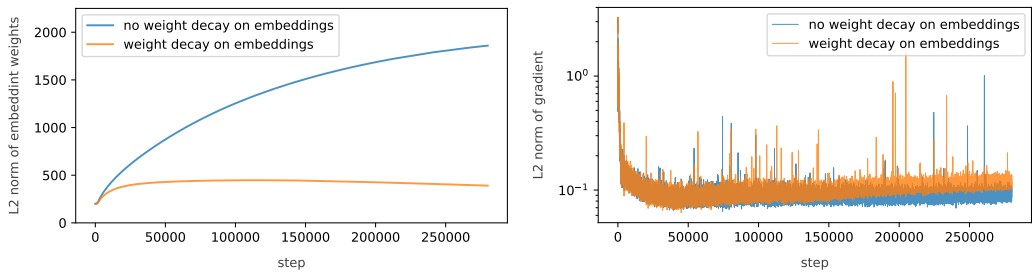

Figure 11: Weight decay applied to token embeddings leads to a gradual decrease in the embedding norm and a corresponding increase in the gradient norm. Decaying embeddings also has a modest negative impact on stability, producing more spikes than a comparable run without (spike scores of 0.16 and 0.092 respectively).

### F.5   Studying the impact of learning rate

Our starting point for learning rate experiments was the setting from Grattafiori et al. (2024). To initialize the optimizer state for the 7B variant, we linearly warm up the learning rate to its peak of $3 \cdot 10E{-}4$ over the first 2000 steps. Then, we use a standard cosine decay over 5T tokens. In OLMO-0424 (Ai2, 2024), we suggested that the last part of a cosine decay schedule can be cut off and replaced by a linear decay to zero with little loss of performance. Accordingly, for the 7B variant, we stop the schedule at 4T tokens and then switch to mid-training as described in Section §5. The 13B ran with a higher peak learning rate from the start, so we decided to run it to 5T tokens before moving to the mid-training stage.

Figure 12 shows different runs with four additional learning rate values: $6 \cdot 10E-4$, $9 \cdot 10E-4$, $12 \cdot 10E-4$, and $30 \cdot 10E-4$. In particular, we tried double, triple, quadruple, 10×, and 30× the original learning rate. The last, $30 \cdot 10E-4$, showed training instabilities already during learning rate warm-up, with several loss spikes that did not recover fully, so we abandoned this variant quickly. The other values trained normally and showed an interesting pattern. Looking purely at training loss, higher learning rates universally perform better early on (as long as they avoid loss spikes), but eventually the lower learning rate setting overtakes the others (Figure 12). Notably, when comparing $3 \cdot 10E-4$ and $6 \cdot 10E-4$, the cross-over point is well past 200B tokens. A shorter hyperparameter experiment might come to the wrong conclusion.

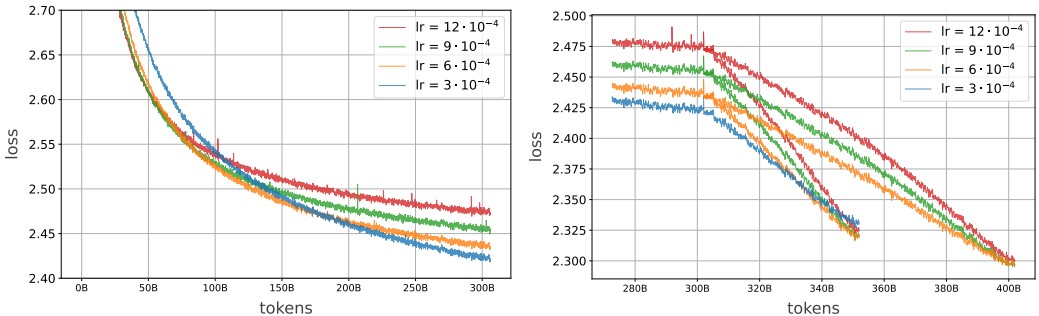

Figure 12: Higher learning rates perform better at first but are eventually overtaken by lower rates. However, linearly decaying the learning rate to zero over 50B or 100B tokens results in equivalent training loss.

One of the motivations for this line of experimentation was to find out whether a higher learning rate would make the annealing step more effective. The conjecture is that the worse training loss during pretraining is compensated for when the learning rate decays to zero. To test this hypothesis, we took a checkpoint from each of our four variants after 300B tokens, and decayed the learning rate to zero over 50B tokens. To account for the possibility that the effect of higher learning rates needs more steps to unfold, we tried the three higher settings and decayed the learning rate over 100B tokens, for a total of seven experiments. The results show that a higher learning rate does make mid-training more effective, but it does so by exactly the amount that the pretraining is worse. All four variants show the same training loss at the end of the procedure, though the lowest setting lags behind the others by a small amount.

Table 12 shows that the result is consistent for longer training runs as well. We took two variants, $3 \cdot 10E-4$ and $6 \cdot 10E-4$, and repeated the experiment after training for 1T and for 2T tokens. We chose these variants because $3 \cdot 10E-4$ is the baseline from Grattafiori et al. (2024), and $6 \cdot 10E-4$ showed, by a slim margin, the best training loss. Our results show virtually no difference between the two settings, both on training loss and a mix of nine downstream tasks from the OLMES suite shown in Table 12. Evaluating the models on downstream tasks is noisier, but mirrors the findings based on training loss only.

Finally, we wanted to see if a higher learning rate during the pretraining stage would result in a more effective mid-training stage when switching to higher quality data. To match our training setup as much as possible within the available compute budget, we took the same two settings ($3 \cdot 10E-4$ and $6 \cdot 10E-4$), and linearly decayed the learning rate to 0 over 100B high quality tokens. Once again, the results show little difference. The final scores on the OLMES evaluation suite are within 0.1 points of each other. However, looking at other metrics may still reveal a meaningful difference between the two settings. The mix of high quality tokens targets math specifically, and on GSM8K (which is not part of the OLMES suite), the high learning rate setting is 2.8 points better than the lower learning rate. More study is needed to turn this interesting data point into a dependable result.

This finding contradicts machine learning folk wisdoms such as "higher learning rates are always better" or "area under the learning curve matters" (McCandlish et al., 2018).

| Learning Rate | Pretraining Stage | Mid-training Stage | OLMES (CF, valid) |
|---|---|---|---|
| $3 \cdot 10E{-}4$ | 300B tokens | 50B tokens | 62.5 |
| $6 \cdot 10E{-}4$ | 300B tokens | 50B tokens | 63.9 |
| $9 \cdot 10E{-}4$ | 300B tokens | 50B tokens | 64.1 |
| $12 \cdot 10E{-}4$ | 300B tokens | 50B tokens | 63.6 |
| $6 \cdot 10E{-}4$ | 300B tokens | 100B tokens | 64.6 |
| $9 \cdot 10E{-}4$ | 300B tokens | 100B tokens | 64.5 |
| $12 \cdot 10E{-}4$ | 300B tokens | 100B tokens | 64.2 |
| $3 \cdot 10E{-}4$ | 2T tokens | 100B high quality tokens | 73.8 |
| $6 \cdot 10E{-}4$ | 2T tokens | 100B high quality tokens | 73.9 |

Table 12: Results on 9 multiple-choice tasks from the *validation* subset of OLMES (*cloze formulation* format) for various peak learning rates and schedule lengths. Average scores vary by less than two points across all variants, with most scores within half a point of each other.

It expands on Wortsman et al. (2023), who observed that smaller models' performance is largely invariant to learning rate over several orders of magnitude when trained to the end of a cosine schedule, and further found that QK-norm (§F.3.1) and z-loss (§F.3.2), which we use as well, enhance this effect. We find that these results still hold even at much larger scales of tokens and parameters, and, crucially for our training efforts, with our modified learning rate schedule.

Due to cost concerns we did not explore the full range of learning rates. This is the main limitation of this line of experimentation. It would be interesting to run a wider sweep of learning rates to accurately define the boundaries of the plateau we appear to be training in.

# G  DOLMINO MIX 1124

## G.1  DOLMINO MIX 1124 High Quality Sources

We start by curating a higher quality subset of Stage 1 data, and expand it with more academic and encyclopedic material. In particular, we consider the following sources:

**High Quality Web:** To filter the web subset used in pretraining, we experiment with two existing quality classifiers:

- **FastText classifier from Li et al. (2024).** To train this model[10], Li et al. sampled positive documents from the Reddit subset in ELI5 (Fan et al., 2019), and demonstrations from Open Hermes 2.5[11]. Negatives are sampled at random from the DCLM pipeline.

- **FineWeb Edu classifier from Penedo et al. (2024).** This model[12] is fine-tuned from the Arctic Embed M[13] encoder (Merrick et al., 2024) on over 400,000 web pages[14] labeled by Llama 3 70B Instruct. This classifier scores documents from 0 to 5 according to adherence to academic topics and polished content.

Following Li et al. (2024), we use the DCLM FastText classifier with a threshold of 0.03311014, which retains approximately 65.6% of the web subset. We combine this filter with the the

---

[10] 🤗 mlfoundations/fasttext-oh-eli5

[11] 🗄 datasets/teknium/OpenHermes-2.5

[12] 🤗 HuggingFaceFW/fineweb-edu-classifier

[13] 🤗 Snowflake/snowflake-arctic-embed-m

[14] 🗄 datasets/HuggingFaceFW/fineweb-edu-llama3-annotations

| | Source | Mix % | | | | | | |
|---|---|---|---|---|---|---|---|---|
| | | PT Mix | Web $^{FT_7}$ | Web $^{FT_7}_{FW_3}$ | Web $^{FT_7}_{FW_2}$ | Web $^{FT_7}_{FW_2}$ + Math | Web $^{FT_7}_{FW_2}$ + Ins | Web $^{FT_7}_{FW_2}$ + Math + Ins |
| | DCLM | 95.2 | 57.1 | 54.2 | 57.9 | 61.8 | 75.5 | 57.5 |
| INST | Flan | - | - | - | - | - | 8.8 | 6.7 |
| | Stack Exchange | - | - | - | - | - | 0.7 | 0.5 |
| CODE | Starcoder | 2.1 | 19.5 | 20.9 | 19.2 | - | - | - |
| | CodeSearchNet | - | - | - | - | 0.1 | 0.2 | 0.1 |
| REFERENCE | Gutenberg Books | - | 1.2 | 1.3 | 1.2 | - | - | - |
| | peS2o | 1.5 | 6.6 | 7.1 | 6.5 | 10.7 | 13.0 | 9.9 |
| | Wikipedia | 0.1 | 0.9 | 0.9 | 0.9 | 1.6 | 1.9 | 1.4 |
| | StackExchange | - | 4.0 | 4.3 | 4.0 | - | - | - |
| | ArXiv | 0.5 | 4.9 | 5.2 | 4.8 | - | - | - |
| MATH | Algebraic Stack | 0.3 | 2.8 | 3.0 | 2.7 | - | - | - |
| | OpenWebMath | 0.3 | 2.9 | 3.1 | 2.8 | 5.2 | - | 4.8 |
| | GSM8k | - | - | 0.003 | 0.003 | 0.003 | - | 0.003 |
| | Mathpile | - | - | - | - | 2.1 | - | 1.9 |
| | AutoMathText | - | - | - | - | 18.5 | - | 17.2 |

Table 13: A summary of high-quality sources we evaluate for mid-training. We experiment with mixing these sources in 6 mixes, each consisting of 50 billion tokens. Percentages on the table indicate the fraction of each 50B mix that is comprised by data from the respective source. PT Mix is sampled (with repetition) from the pretraining stage.

scores from FineWeb Edu classifier; we experiment by retaining documents with score over 3 (5.8% retained), as well as a more relaxed threshold of 2 (20.3% retained).

**Instruction data and Q&A pairs** We leverage the same subset of FLAN Wei et al. (2021); Longpre et al. (2023) from DOLMA 1.7 (Soldaini et al., 2024). We decontaminated this source by extracting training, validation, and test instances from all tasks in our evaluation suite (Section §3) and removed FLAN documents with 10% or more overlapping ngrams with any task instance.

We source question and answer pairs from the Stack Exchange network, a collection of 186 forums dedicated to a wide variety of topics. Content on Stack Exchange network is licensed under various commercial-friendly Creative Common licenses. We use the latest database dump (September 30[th], 2024) at the time of writing, which is distributed by the Internet Archive[15]. We filter questions to those that have an accepted answer; further, we Q&A pairs whose questions have fewer than 3 votes or answers have fewer than 5 votes. Once filtered, we concatenate questions and answers together using a sequence of new lines that contains one more \n than longest sequence of newlines in either the question or answer.

**Code** We evaluate retaining the same subset of code used during pretraining; furthermore, we consider smaller, curated sources of code interleaved with natural supervision, such as docstrings in CodeSearchNet (Husain et al., 2019); Q&A pairs from StackExchange described in the paragraph above also contain code.

**Academic, encyclopedic and other reference content** We source high-quality non-web datasets from Dolma 1.7 (Soldaini et al., 2024). This includes peS2o (Soldaini & Lo, 2023), Wikipedia, and Wikibooks, Gutenberg books, arXiv and StackExchange (from Red-Pajama v1; Together AI, 2023), Algebraic Stack (ProofPile II; Azerbayev et al., 2023).

**Math** In parallel to developing the math subset of DOLMINO MIX 1124, we consider preliminary math subset to gauge how math documents combine with the non-math portion of the mix. In particular, we used OpenWebMath (Paster et al., 2023), the train split of

---

[15]archive.org/details/stackexchange_20240930

GSM8K (Cobbe et al., 2021), the train split of the permissively licensed ("commercial") subset of MathPile (Wang et al., 2023), and AutoMathText (Zhang et al., 2024b).

# H DOLMINO MIX 1124 Subsamples

To perform model souping, we subsample DOLMINO MIX 1124 to create 50B, 100B, and 300B subsets. We perform mid-training on each to realize the model soup ingredient checkpoints. We summarize these subsamples in Table 14.

| Source | Tokens | 50B | | 100B | | 300B | |
|---|---|---|---|---|---|---|---|
| | | Source % | Mix % | Source % | Mix % | Source % | Mix % |
| Filtered DCLM | 752B | 3.23 | 47.2 | 6.85 | 50.2 | 20.78 | 51.9 |
| Decontam. FLAN | 17.0B | 50.0 | 16.6 | 100 | 16.7 | 200 | 11.3 |
| StackExchange Q&A | 1.26B | 100 | 2.45 | 200 | 2.47 | 400 | 1.68 |
| peS2o | 58.6B | 5.15 | 5.85 | 16.7 | 9.52 | 100 | 19.4 |
| Wikipedia/Wikibooks | 3.7B | 100 | 7.11 | 100 | 3.57 | 400 | 4.86 |
| DOLMINO MIX 1124 Math Mix | 10.7B | 100 | 20.8 | 200 | 17.5 | 400 | 10.8 |

Table 14: DOLMINO MIX 1124 compositions. The Source % column indicates the fraction of the source that was used in the DOLMINO MIX 1124 mix. Numbers in this column greater than 100 indicate we used the data, e.g. 400 indicates a 4x repeat. The Mix % column describes the proportion of the DOLMINO MIX 1124 mix that is composed of this source, i.e., this column should sum to 100%.

# I Difficulties with OLMO 2 1B

We developed our OLMO 2 recipe developed using the OLMO 2 1B model (Appendix B) and have found findings to generalize well to the 7B, 13B and 32B scales, as seen by our competitive results in Table 3. Yet, we have found scaling the number of training tokens for OLMO 2 1B to be difficult.

**Training** We pretrain OLMO 2 1B to 4 trillion tokens on OLMO 2 MIX 1124 and perform a single 50B token anneal on DOLMINO MIX 1124. Similar to OLMO 2 7B, we use 2000 steps of warmup, set the schedule to 5 trillion tokens but truncate at the 4 trillion mark. We use a higher peak learning rate of $4.0 \cdot 10E-4$.

**Base Results** Table 15 presents experimental results on our main base model evaluation suite. We find that while OLMO 2 remains competitive with other similarly-sized models like SmolLM 2, it lags behind the smaller Gemma 2 and Qwen 2.5 base models.

| Model | Avg | FLOPs | MMLU | ARC$_C$ | HS | WG | NQ | DROP | AGI | GSM | MMLU$_P$ | TQA |
|---|---|---|---|---|---|---|---|---|---|---|---|---|
| | | | **Dev Benchmarks** | | | | | | **Held-out Evals** | | | |
| | | | **Open-weights models 1-2B Parameters** | | | | | | | | | |
| Qwen 2.5 1.5B | 51.5 | 1.7 | 61.4 | 77.3 | 67.0 | 65.4 | 17.7 | 36.4 | 47.9 | 63.2 | 29.9 | 49.1 |
| Gemma 2 2B | 47.9 | 0.2 | 53.1 | 67.4 | 74.4 | 70.8 | 24.1 | 36.9 | 38.4 | 26.8 | 22.2 | 65.2 |
| | | | **Fully-open models** | | | | | | | | | |
| SmolLM 2 1.7B | 44.7 | 1.1 | 50.9 | 62.0 | 73.3 | 66.9 | 19.1 | 26.5 | 35.3 | 30.3 | 22.0 | 60.6 |
| OLMO 2 1B | 43.7 | 0.4 | 44.3 | 51.3 | 69.5 | 66.5 | 20.8 | 34.0 | 36.3 | 43.8 | 16.1 | 54.7 |

Table 15: OLMO 2 1B vs. comparable models (size, architecture) with known pretraining FLOPs (relative to 10E23).

**Analysis** We postulate that our OLMO 2 1B may struggle with pretraining token efficiency due to model capacity. OLMO 2 is smaller than the smallest variants of other competitive

| Model | Avg | AE2 | BBH | DROP | GSM | IFE | MATH | MMLU | Safety | PQA | TQA |
|---|---|---|---|---|---|---|---|---|---|---|---|
| **Open weights models 1–2B Parameters** | | | | | | | | | | | |
| Gemma 3 1B | 38.3 | 20.4 | 39.4 | 25.1 | 35.0 | 60.6 | 40.3 | 38.9 | 70.2 | 9.6 | 43.8 |
| Llama 3.2 1B | 39.3 | 10.1 | 40.2 | 32.2 | 45.4 | 54.0 | 21.6 | 46.7 | 87.2 | 13.8 | 41.5 |
| Qwen 2.5 1.5B | 41.7 | 7.4 | 45.8 | 13.4 | 66.2 | 44.2 | 40.6 | 59.7 | 77.6 | 15.5 | 46.5 |
| **Fully-open models** | | | | | | | | | | | |
| SmolLM2 1.7B | 34.2 | 5.8 | 39.8 | 30.9 | 45.3 | 51.6 | 20.3 | 34.3 | 52.4 | 16.4 | 45.3 |
| OLMO 2 1B | 42.7 | 9.1 | 35.0 | 34.6 | 68.3 | 70.1 | 20.7 | 40.0 | 87.6 | 12.9 | 48.7 |

Table 16: OLMO 2-INSTRUCT 1B's performance vs open-weights models of comparable size.

model families like Qwen 2.5 or Gemma 2. We hypothesize that below a certain model size, the optimal pretraining recipe may require the inclusion of task-specific data, such as that seen in supervised fine-tuning (SFT) to achieve non-random performance over more challenging tasks in our evaluation suite. Better performance could also be achieved by distilling from a more powerful model, a strategy used by the smaller Gemma 2 models.

For example, Table 17 shows the benefit of DOLMINO MIX 1124 is higher with smaller base models: +37.0% for the 1B model, +18.7% for the 7B model, +15.9% for the 13B model, and +12.3% for the 32B model. These results also show that OLMO 2 1B with only Stage 1 pretraining struggles to break out of random performance for multiple-choice formatted tasks (25% for MMLU and ARC Challenge, 10% for MMLU Pro).

As further evidence of this, Table 16 shows that applying our same OLMO 2-INSTRUCT post-training recipe to OLMO 2 1B results in OLMO 2-INSTRUCT 1B with highly competitive performance to even Qwen 2.5 and even Gemma 3.

| | | | Dev Benchmarks | | | | | | Held-out Evals | | | |
|---|---|---|---|---|---|---|---|---|---|---|---|---|
| Model | Stage | Avg | MMLU | ARC$_C$ | HS | WG | NQ | DROP | AGI | GSM | MMLU$_P$ | TQA |
| 1B | 1 | 31.9 | 26.9 | 26.1 | 67.5 | 67.8 | 16.1 | 25.1 | 24.5 | 3.3 | 11.1 | 50.1 |
| | 2 | 43.7 | 44.3 | 51.3 | 69.5 | 66.5 | 20.8 | 34.0 | 36.3 | 43.8 | 16.1 | 54.7 |
| 7B | 1 | 53.0 | 59.8 | 72.6 | 81.3 | 75.8 | 29.0 | 40.7 | 44.6 | 24.1 | 27.4 | 74.6 |
| | 2 | 62.9 | 63.7 | 79.8 | 83.8 | 77.2 | 36.9 | 60.8 | 50.4 | 67.5 | 31.0 | 78.0 |
| 13B | 1 | 58.9 | 63.4 | 80.2 | 84.8 | 79.4 | 34.6 | 49.6 | 48.2 | 37.3 | 31.2 | 80.3 |
| | 2 | 68.3 | 67.5 | 83.5 | 86.4 | 81.5 | 46.7 | 70.7 | 54.2 | 75.1 | 35.1 | 81.9 |
| 32B | 1 | 64.9 | 72.9 | 88.7 | 86.5 | 82.4 | 40.6 | 57.3 | 56.8 | 56.2 | 42.0 | 85.5 |
| | 2 | 72.9 | 74.9 | 90.4 | 89.7 | 83.0 | 50.2 | 74.3 | 61.0 | 78.8 | 46.9 | 88.0 |

Table 17: OLMO 2 1B requires our mid-training recipe to break out of near-random performance on multiple-choice tasks like MMLU, ARC Challenge, and MMLU Pro.

## J   Comparison with Qwen 3 Base

Concurrent with this work is Qwen 3 (Yang et al., 2025). Table 18 presents evaluation results for Qwen 3 base model, following the presentation used in Table 3. We omit pretraining FLOPs as Qwen 3's use of mixture-of-experts is not directly comparable to OLMO 2 autoregressive model family. We find that Qwen 3's base model overall performs similarly to Qwen 2.5's base model but Qwen 3 takes some design decisions (possibly to support newer research threads around reasoning) that may be incompatible with performing well on base model evaluations (see drop in GSM8k performance likely due to format mismatch). This motivates future work for us to refine our base model evaluation suite to reflect these growing trends in language model research.

| Model | Avg | MMLU | ARC$_C$ | HS | WG | NQ | DROP | AGI | GSM | MMLU+ | TQA |
|---|---|---|---|---|---|---|---|---|---|---|---|
| | | Dev Benchmarks | | | | | | Held-out Evals | | | |
| Qwen 2.5 7B | 67.4 | 74.4 | 89.5 | 89.7 | 74.2 | 29.9 | 55.8 | 63.7 | 81.5 | 45.8 | 69.4 |
| Qwen 3 8B | 66.6 | 76.8 | 91.2 | 89.5 | 69.9 | 21.8 | 61.8 | 64.3 | 74.8 | 50.6 | 66.5 |
| OLMo 2 7B | 62.9 | 63.7 | 79.8 | 83.8 | 77.2 | 36.9 | 60.9 | 50.4 | 67.5 | 31.0 | 78.0 |
| Qwen 2.5 14B | 72.3 | 79.3 | 94.0 | 94.0 | 80.0 | 37.3 | 51.5 | 71.0 | 83.4 | 52.8 | 79.2 |
| Qwen 3 14B | 73.6 | 80.7 | 93.4 | 92.3 | 76.4 | 31.8 | 75.0 | 70.3 | 87.3 | 55.7 | 73.2 |
| OLMo 2 13B | 68.3 | 67.5 | 83.5 | 86.4 | 81.5 | 46.7 | 70.7 | 54.2 | 75.1 | 35.1 | 81.9 |
| Qwen 2.5 32B | 74.9 | 83.1 | 95.6 | 96.0 | 84.0 | 37.0 | 53.1 | 78.0 | 83.3 | 59.0 | 79.9 |
| Qwen 3 32B | 68.9 | 83.3 | 94.9 | 93.5 | 79.0 | 31.9 | 67.4 | 72.4 | 34.0 | 60.7 | 72.2 |
| OLMo 2 32B | 73.3 | 74.9 | 90.4 | 89.7 | 78.7 | 50.2 | 74.3 | 61.0 | 78.8 | 46.9 | 88.0 |

Table 18: OLMo 2 vs Qwen base models.

