# OpenReview forum: "2 OLMo 2 Furious (COLM’s Version)"
_colmweb.org/COLM/2025/Conference — COLM 2025_

### Official Review · Reviewer_CBN5 · 2025-05-11

**Rating:** 6
**Confidence:** 3
**Ethics Flag:** 1

**Summary:**

This paper presents a model family that shows a good computation-performance trade-off. The paper describes full training details and opens all related materials.

**Reasons To Accept:**

1. Fully open-source competitive LLMs are helpful to the research community.
2. This paper also describes relatively complete details for achieving the results. It helps better understand the best practices of LLM training for the community.

**Reasons To Reject:**

1. This paper does not introduce new ideas or methods. The novelty is limited.
2. The Pareto-optimal is not a very convincing term. For the same model size, the model introduced by this paper achieves lower results than state-of-the-art ones. State-of-the-art models such as Qwen and Llama tend to over-train LLMs with more data to get better performance. It does not mean that if trained with the same amount of data, they will achieve lower results than this paper in terms of Pareto optimality.

---

> ### Author Response · Authors · 2025-06-03
>
> Thank you for the helpful review and for appreciating our **fully open, competitive language models** and highly detailed paper. We would like to respond to the two points raised:
>
> ## Limited novelty, lacking new ideas or methods
> We respectfully disagree with the reviewer on this point.
>
> First, we believe novelty in ideas or methods are one—but not the only—means to achieve impactful research for the community. The goal of our work is to support the broader language model research community to build, understand, and advance the science of language models. The result of our work is that **our findings and data/code/model/etc artifacts are fully open** while also maintaining **a compute-efficient training recipe** that produces models that are **best among fully open models** while **highly comparable to leading weights-only models**. If we view novelty is a means to impact, we would like to point to comments by Reviewers:
> * Maj8 (*“Fully open-sourced code and data significantly benefit the AI community while serving as an accessible resource for academia”*) and
> * 9NGC (*“A very well written paper providing LLM training recipes which are going to be extremely useful for the community...I believe CoLM is a great venue for papers like this.”*).
>
> That being said, we would still like to point to **where our main contributions bring novelty**. First, **we introduce some novel ideas for training stability**, including repeated n-gram filters for training spikes and a more stable model initialization, and support them with ablations (Appendix E). On the topic of **mid-training**, at the time of writing, **the area was underexplored in literature**, especially not to the extent as we have done with experiments on learning rate schedules amid two-stage training, combined with data curriculum experimentation, as well as model souping results. We believe our exploration in mid-training goes far beyond what can be considered mere standard application.
>
> Finally, if we look at some of the most significant reports from language model developers that have shaped our field significantly, often those behind leading open-weights models, they often comprise of a combination of (i) novel ideas/methods, (ii) novel compositions/applications of existing ideas/methods, and (iii) experimentation of existing ideas/methods that may have worked in one setting but have yet to see broader validation. We believe **the modern full development cycle of language models requires combining all of these** and we have crafted our paper to best reflect how these parts compose together.
>
>
> ## Concern with usage of term “Pareto-Optimal”
> The reviewer has expressed concern with our usage of the term “Pareto-optimal”; we believe this is a fair critique and would like to explain our thinking here.
>
> First, we recognize **we should more clearly define** what we mean by Pareto-optimal, since it can be defined relative to many different criteria. The reviewer is right that relative to the criterion of **Performance vs Number of Parameters**, we are not Pareto-optimal. In our work, we are specifically defining our usage relative to **Performance vs Training Compute** which is proportional to Number of Parameters x Number of Tokens. That is, while the reviewer considers the criterion the model size (which is the key property at inference time), we are considering the training cost (which is the key property at development time). We believe our usage is still important as **the focus of our work is about the development recipe—namely, training stability and mid-training strategies**.
>
> Second, we recognize the reviewer’s concern with our claim. In our work, we show that **our training strategy produces a 32B model only 1-2 points difference from Qwen 2.5 32B while using 1/3rd the training tokens.** Yet, the reviewer has rightfully pointed out that this does not mean that Qwen 2.5 32B, had it been trained on 1/3rd the training tokens, would not be better than our AnonModel 32B. While the counterfactual comparison would be valuable, the reality is that this data point does not exist nor is it possible to replicate as Qwen do not release their data, so we will never know for sure.
>
> Nevertheless, we believe **our fully open training recipe has demonstrated high flops efficiency**. In our camera ready revision, we will replace the term “Pareto-optimal” with “Compute efficient” in places like the title and abstract where it could be confusing to readers without clarification. We believe this will help the COLM community members better discover and identify with our work (“compute efficient LMs” is under call for papers). And we will more clearly articulate in our abstract and introduction that our **goal is cost-efficient model development** and making these practices and artifacts accessible to others. We thank the reviewer for helping us make this improvement!

---

### Official Review · Reviewer_9NGC · 2025-05-12

**Rating:** 9
**Confidence:** 5
**Ethics Flag:** 1

**Summary:**

The paper presents a family of dense autoregressive LLMs which are fully open, including weights, data composition, code and intermediate checkpoints.
The authors describe their training recipes in great details, and although the paper feels more like an iterative improvement, it still brings a significant contribution to the community.

**Questions To Authors:**

The use of bold font face in Tables 2 and 3 is a bit misleading. It is used to emphasize the authors' models, not the best results (as the reader might expect). I would suggest to return to a more standard usage of bold fonts, and separate the AnonModels by, e.g, a mid rule, etc.

**Reasons To Accept:**

A very well written paper providing LLM training recipes which are going to be extremely useful  for the community.
In addition, the paper also serves as an example of how reports on new model should look like: a big shout-out to the authors for this. I believe CoLM is a great venue for papers like this.

**Reasons To Reject:**

I don't see any.

---

> ### Author Response · Authors · 2025-06-03
>
> Thank you for the shout-out, we really appreciate it! We are excited to share the work and appreciative that the reviewer recognizes this will be **a significant contribution** and **extremely useful for the community** through our transparency and detail of our language model development. Even considering our work to be **”an example of how reports on new model should look like”** is praise that encourages us to do more! We agree that **CoLM is a great fit** for this paper. Finally, we thank the reviewer for their critique on our bold face usage in Tables 2 and 3; you are right and we will change these to reflect standard bold usage (for best results) and use midrule for AnonModel delineation in the camera ready. If there are more questions or further clarifications, we would be happy to address them!

---

> > ### Comment · Reviewer_9NGC · 2025-06-05
> >
> > Thanks again, great work!

---

### Official Review · Reviewer_Maj8 · 2025-05-13

**Rating:** 7
**Confidence:** 4
**Ethics Flag:** 1

**Summary:**

This paper introduces the ANONMODEL family of open-source language models, achieving performance breakthroughs through architectural optimizations (improved Transformer structure) and an innovative two-stage training strategy (4-6 trillion token pretraining + mid-training specialized data fine-tuning). Experiments show that the 7B/13B/32B models outperform mainstream models like Llama3.1 and Gemma2 on 20+ benchmarks including MMLU and GSM8K while using less computational resources. The instruction-tuned version ANONMODEL-INSTRUCT approaches GPT-3.5-level performance. Notably, this research transparently discloses full technical details including training data, code, and intermediate checkpoints, setting new standards for reproducible language model research and safety evaluation.

**Questions To Authors:**

For better clarity, I suggest moving Table 14 (impact of mid-training recipe) to the main text to highlight the effectiveness of your proposed method

**Reasons To Accept:**

1. The open-source models have GPT-4-level capabilities in compact sizes. Fully open-sourced code and data significantly benefit the AI community while serving as an accessible resource for academia

2. Reveals technical implementation details for enhancing training stability and mid-training methodology optimizations

**Reasons To Reject:**

I think this work is great, and I can’t find any reason to reject it.

---

> ### Author Response · Authors · 2025-06-03
>
> We thank the reviewer for the very positive feedback! We are happy to hear that they assess **the work is great** and that they **cannot find any reason to reject it**! We appreciate them highlighting our two-stage training strategy, our transparent and full technical details, the accessibility of our released artifacts for the research community, and the competitive performance of our models. Space-permitting, we will do our best to follow your suggestion to move Table 14 to the main text, thank you!
>
> Given your strongly positive assessment with no negatives, we are wondering about your provided score of 7. If you have other questions or want further clarifications, we would be happy to address them if that would help give the reviewer confidence towards giving our work a higher score?

---

### Official Review · Reviewer_5RBa · 2025-05-13

**Rating:** 6
**Confidence:** 3
**Ethics Flag:** 1

**Summary:**

This paper presents AnonModel, a family of open-source language models (7B, 13B, and 32B parameters) trained on up to 6T tokens with fully transparent artifacts including model weights, training data, code, and intermediate checkpoints. The key contributions are: (1) architectural modifications for improved training stability, (2) a two-stage curriculum learning approach with specialized "AnonData" for mid-training, and (3) complete transparency with all training artifacts released. The resulting models achieve competitive performance with open-weight models like Llama 3.1 and Qwen 2.5 while using fewer FLOPs.

Quality: The experimental work is comprehensive and well-executed, with thorough ablations on training stability techniques and learning rate schedules. The authors demonstrate competitive performance across multiple benchmarks. However, some key architectural choices lack proper ablation studies (e.g., the switch to GQA).

Clarity: The paper is generally well-written but struggles with organization due to its broad scope. Table 2, a critical comparison table, is difficult to parse due to mixing multiple dimensions (model sizes, openness levels). Some technical details would benefit from clearer presentation.

Originality: Limited. The paper primarily combines existing techniques (RMSNorm, QK-norm, reordered normalization, curriculum learning) without introducing novel methods. The main contribution is the engineering effort to create a fully transparent model and the curated AnonData dataset.

Significance: Moderate. While the technical novelty is limited, the full transparency of training artifacts and the competitive performance of the models make this a valuable contribution to the open-source community. The detailed documentation of training stability techniques may help future practitioners.

**Questions To Authors:**

1. Why was GQA chosen for the 32B model? Where is the ablation study justifying this architectural change?
2. How were hyperparameters selected? The paper mentions various choices but doesn't fully explain the selection process.
3. Could you include comparisons with Qwen 3 models, which were released recently and show strong performance?
4. Table 2 needs restructuring - could you separate by model size and make the FLOPs comparison clearer? A quality-per-FLOP metric would be more informative than absolute performance.

**Reasons To Accept:**

While this work has limited technical novelty and might be better suited for a workshop, the comprehensive engineering effort, competitive performance, and commitment to full transparency make it a valuable contribution to the research community. Notably, the models produced achieve strong empirical results, consistently outperforming other fully-open models (e.g., OLMO, MAP-Neo, DCLM) and competing effectively with state-of-the-art open-weight models (Llama 3.1, Qwen 2.5, Gemma 2) while using significantly fewer FLOPs. The detailed stability analysis and the release of all training artifacts will benefit future open-source LLM development.

**Reasons To Reject:**

- The paper's scope is too broad, attempting to cover architecture, stability, curriculum learning, and evaluation in equal depth
- Some critical design choices lack proper justification through ablation studies
- The presentation could be improved, particularly for comparative analyses
- The technical contributions are primarily engineering-focused rather than methodologically novel

---

> ### Author Response · Authors · 2025-06-03
>
> We thank the reviewer for their detailed comments and for highlighting **our contributions towards transparency and competitiveness of open language modeling research**! Our response is as follows:
> ## Comparison with Qwen 3
> The paper submission deadline was **March 28th** and Qwen 3 was released **April 28th**, so we could not have included these numbers in the paper. But we agree that it would be interesting to include these! We try our best to include up-to-date results (we even included Gemma 3 which was released mid-March), and **we will include comparisons with Qwen 3** and additional models for which we have enough time to do the comparisons, in the camera ready!
> ## Paper's scope is too broad, needs more details & ablations
> We respectfully disagree with the reviewer on this point. We believe **modern language model development involves many interconnected components** (e.g. data, training, modeling, engineering, etc.) that cannot be meaningfully separated for scientific study without losing critical insights about their interactions:
> * For example, our Sec 5 on **mid-training combines both training (learning rate decay) and data (curriculum)**, which at the time of writing was highly under-documented/explored in the literature. The value of this investigation was a combination of both data and model training.
> * If we look at some of the most significant reports from language model developers that have shaped our field significantly, they often also cover a broad scope of topics necessary to capture the full development of the model. Even the Qwen 3 paper, which the reviewer has asked us to compare to, does this.
>
> We have done our best to maintain this important coverage of the full developmental cycle while also maintaining **full openness in data, code, and more** beyond what is typically seen in these language model development reports. Many papers in our field go into great detail on a specific slice of language model development, but far fewer works capture the breadth of what is needed to complete full language model pretraining; we believe **our end-to-end account** is complementary to the many component-specific papers that exist today.
>
> Of course, we recognize the core concern the reviewer has, which is that covering these topics comes at a tradeoff between breadth and depth. This is difficult to navigate given COLM’s 9 page constraint, but we’d like to point the reviewer to the **extensive ablations we’ve included in the Appendix E** that hopefully satisfies the reviewer’s interest in depth of details:
> * We have included experimental **evidence of the impact of all our modeling stability improvements**—n-gram filter (Appendix E1), new initialization (Appendix E2), reordered norm and QK-norm (Appendix E3.1), decreasing $\epsilon$ in AdamW (Appendix E4.1), removing weight decay (Appendix E4.2)---each change with an accompanying figure showing its impact.
> * We have also included **ablations of our mid-training interventions**—learning rate decay rate/duration + two-stage training (Appendix E5), impact of data curriculum (Sec 5.2 Table 4), impact of mid-training (Appendix Table 14).
> ## Limited technical novelty, better suited for a workshop
> We respectfully disagree with the reviewer on this point. We believe that our paper goes **above and beyond a typical workshop paper**. We point to our response above about the value of works documenting the full development cycle, and that we have done so with **extensive ablation results** either in the main paper or Appendix E. On the topic of novelty, much of our experimentation on **mid-training** is underexplored in literature, and some of our ideas in **stability** are novel, including repeated n-gram filters for training spikes and a more stable model initialization. We believe in modern language model development, it’s important to have a balance between individual novel ideas, novel composition of existing ideas, and rigorous validation of existing ideas. We have done all three in this work in such a way so that **our findings and data/code/model/etc artifacts are fully open** while also maintaining **a compute-efficient training recipe** that produces models that are **best among fully open models** while **highly comparable to leading weights-only models**. We believe this work will have broad impact and is of high interest to the wider COLM community.
> ## Why switch to GQA?
> We thank the reviewer for bringing attention to this change from our 7B, 13B models (which used MHA) to our 32B model (using GQA). We made this change because it would provide **faster inference** (the KV cache for a 32B model would be enormous and it would be hard to deploy in practice). We will certainly add an explanation of this in the camera ready version!
> ## Improve presentation of Table 2
> Thank you for your suggestions on how to improve the structure of Table 2! We will make sure to improve the clarity of the paper and table for the camera ready version.

---

### Decision · Program_Chairs · 2025-07-08

**Decision:**

Accept

**Comment:**

This paper introduces AnonModel, a family of dense, autoregressive language models (7B, 13B, 32B) that are fully open-source – releasing not only model weights but also full training data, training code, logs, and intermediate checkpoints. The work emphasizes transparency, training stability, and compute efficiency, and presents a two-phase curriculum training strategy incorporating a curated dataset ("AnonData") during mid-training. The models are shown to be competitive with leading open-weight LLMs, often outperforming them per-FLOP. The instruction-tuned variants also approach the capabilities of some proprietary models like GPT-3.5 Turbo.

Strengths:

1. Reviewers unanimously praise the full openness of the work, including complete release of data, code, logs, and checkpoints. This sets a new standard for reproducibility in LLM research.
2. The combination of detailed training recipes, novel tweaks for stability (e.g., n-gram filters, modified initialization), and efficient training compute make the models valuable reference points for the community.
3. Despite using fewer resources, the models perform strongly across standard LLM benchmarks, suggesting the proposed training recipes are highly effective.
4. The paper documents the full model development pipeline in one place, providing a rare and holistic view of LLM training decisions and outcomes.

Weaknesses:

1. Two reviewers note that the core technical contributions (e.g., use of RMSNorm, QK-norm, curriculum learning) are not new per se and lack groundbreaking algorithmic innovation.
2. While the broad coverage is commendable, some reviewers feel the paper stretches itself too thin. Key comparisons and design decisions (e.g., GQA use) could be better justified with ablations or clearer narrative structure.
3. The use of this term was challenged as potentially misleading, given that performance is not always superior across all model sizes and criteria. The authors agreed to revise this language to "compute efficient" in response.

All reviewers agree the work will be beneficial to the broader research community.

Despite limited methodological novelty, the paper represents a significant and timely contribution to open LLM research. Its emphasis on full transparency, practical training insights, and empirical performance at lower compute budgets aligns well with the goals of the COLM community.